| Editor's Pick | Food Microbiology | Research Article

# Spatial metabolomics reveals the role of penicillic acid in cheese-rind microbiome disruption by a spoilage fungus

Carlismari O. Grundmann,[1] Christopher J. Tomo,[2] Julia L. Hershelman,[2] Benjamin E. Wolfe,[2] Laura M. Sanchez[1]

**ABSTRACT** Microbial interactions in cheese rinds influence community structure, food safety, and product quality. But the chemical mechanisms that mediate microbial interactions in cheeses and other fermented foods are generally not known. Here, we investigate how the spoilage mold *Aspergillus westerdijkiae* chemically inhibits beneficial cheese-rind bacteria using a combination of omics technologies. In cheese-rind community and co-culture experiments, *A. westerdijkiae* strongly inhibited most cheese-rind community members. In co-culture with *Staphylococcus equorum*, *A. westerdijkiae* strongly affected bacterial gene expression, including upregulation of a putative *bceAB* gene cluster that is associated with resistance to antimicrobial compounds in other bacteria. Mass spectrometry imaging revealed spatially localized production of secondary metabolites, including penicillic acid and ochratoxin B at the fungal-bacterial interface with *Brachybacterium alimentarium*. Integration of liquid chromatography-tandem mass spectrometry and genome annotations confirmed the presence of additional bioactive metabolites, such as notoamides and circumdatins. Fungal metabolic responses varied by bacterial partner, suggesting species-specific chemical strategies. Notably, penicillic acid levels increased 2.5-fold during interaction with *B. alimentarium*, and experiments with purified penicillic acid showed inhibition in a dose-dependent manner against this rind bacterium. These findings show that *A. westerdijkiae* deploys a context-dependent suite of mycotoxins and other metabolites, disrupting microbial community assembly in cheese rinds.

**IMPORTANCE** This study identifies the chemical mechanisms underlying the negative impacts of *Aspergillus westerdijkiae* on cheese-rind development, revealing how specialized metabolites like penicillic acid and ochratoxin B influence rind bacterial communities. By integrating biosynthetic gene cluster analyses with mass spectrometry, we demonstrate how chemical communication shapes microbial interactions, with possible implications for food safety and cheese quality. Understanding these interactions is essential for assessing the risks of fungal-driven spoilage and mycotoxin production in cheese-rind maturation. Beyond cheese, these findings contribute to broader microbiome ecology, emphasizing how secondary metabolites mediate microbial competition in natural and fermented food environments.

**KEYWORDS** natural rind cheese, *Aspergillus westerdijkiae*, *Brachybacterium*, *Staphylococcus equorum*, fungal-bacterial interaction, imaging mass spectrometry, mycotoxins

**Peer Reviewer** Scott A. Jarmusch, Danmarks Tekniske Universitet, Kongens Lyngby, Denmark

Address correspondence to Laura M. Sanchez, lmsanche@ucsc.edu, or Benjamin E. Wolfe, benjamin.wolfe@tufts.edu.

The authors declare no conflict of interest.

See the funding table on p. 19.

Cheese rinds host microbial communities composed of both bacteria and fungi that form a biofilm as the cheese ages, playing a crucial role in shaping its sensory properties, texture, and overall quality of the final product. The microbial interactions occurring in this environment, ranging from strong positive to strong negative pairwise interactions, influence the assembly of cheese-rind communities (1–4). Many past studies have shown that fungi are major drivers of microbial interactions in cheese rinds,

providing nutrients to co-occurring bacteria as they break down the cheese substrate. While a few studies have identified the metabolites produced by fungi that mediate cheese-rind assembly (5–7), most mechanisms of interactions between fungi and other microbes in cheese rinds have not been identified (8).

The invasion of cheese-rind microbiomes by undesirable spoilage fungi provides unique opportunities to determine how fungal metabolites can mediate microbial community formation in cheese rinds and other microbiomes. A range of spoilage fungal species can be unintentionally introduced to cheese aging environments, including *Penicillium* and *Aspergillus* species. These fungi can negatively affect cheese quality by altering its appearance, introducing off-flavors, and producing secondary metabolites or mycotoxins that pose safety risks (1, 9, 10). One notable example is *Aspergillus westerdijkiae*, a filamentous fungus not typically associated with cheese rinds but occasionally detected in contaminated dairy products (11–13). We have recently observed this fungus invading cheese facilities in New England, where it can disrupt normal cheese-rind development (Benjamin Wolfe, personal observations). Past work has shown that this fungus secretes the mycotoxins ochratoxin A (OTA), but the diversity and spatial distribution of bioactive metabolites produced by *A. westerdijkiae* in food contexts remains unknown (14–16).

The spatial distribution of metabolites plays a critical role in mediating microbial interactions in structured environments like cheese rinds, where chemical signaling could be localized and species-specific (8, 17, 18). In this context, the bacterial-fungal interactions in cheese rinds can be investigated using matrix-assisted laser desorption/ionization mass spectrometry imaging (MALDI-MSI), which enables the direct spatial visualization of metabolite localization within agar-based microbial cultures. MALDI-MSI not only identifies which fungal metabolites are produced, but also reveals where these compounds accumulate in relation to surrounding microbial colonies (19–21). By comparing the spatial distribution of key secondary metabolites across interactions with different bacterial species, we can assess whether *A. westerdijkiae* exhibits species-specific metabolic responses.

To improve the metabolite annotation and interpretation of MALDI-MSI data, integrated approaches that employ fungal genome mining and molecular networking via the GNPS platform can be used (22, 23). Biosynthetic gene cluster (BGC) predictions help identify the genetic potential of fungi to produce specific secondary metabolites, providing a genomic basis for the observed chemical signals (24). Together, these integrated tools create a powerful framework for linking the spatial organization of metabolite production to underlying biosynthetic pathways, offering deeper insights into how fungi such as *A. westerdijkiae* chemically interact with other microbial species.

The specific microbial community modeled in this study is a washed-rind cheese, one type of artisan cheese where *A. westerdijkiae* has been observed directly disrupting rind formation (Fig. 1A). Washed-rind cheeses are repeatedly treated with a brine solution during aging to promote the development of a surface community dominated by bacteria and yeasts. Common bacterial genera include *Staphylococcus*, *Brevibacterium*, *Brachybacterium*, and various Proteobacteria, while *Debaryomyces hansenii* is the most prevalent yeast (4). Molds (filamentous fungi) are typically not abundant in washed rinds due to the frequent washing, although *Fusarium* (or *Bifusarium*) *domesticum* can persist despite this process (3, 4). The presence of *A. westerdijkiae* in washed-rind cheeses leads to the formation of patches where rinds do not develop and the surface of the cheese is white or brown instead of typical orange colors. This lack of typical rind development led us to hypothesize that *A. westerdijkiae* has antimicrobial activity that prevents beneficial rind microbes from growing.

We integrated MALDI-MSI spatial metabolomics with a variety of other approaches to comprehensively understand how *A. westerdijkiae* disrupts washed-rind cheese-rind development. First, we used interaction experiments with a model cheese-rind community to determine what bacterial and fungal species are inhibited by *A. westerdijkiae*. Once we completed this interaction screen, we used transcriptome sequencing on one

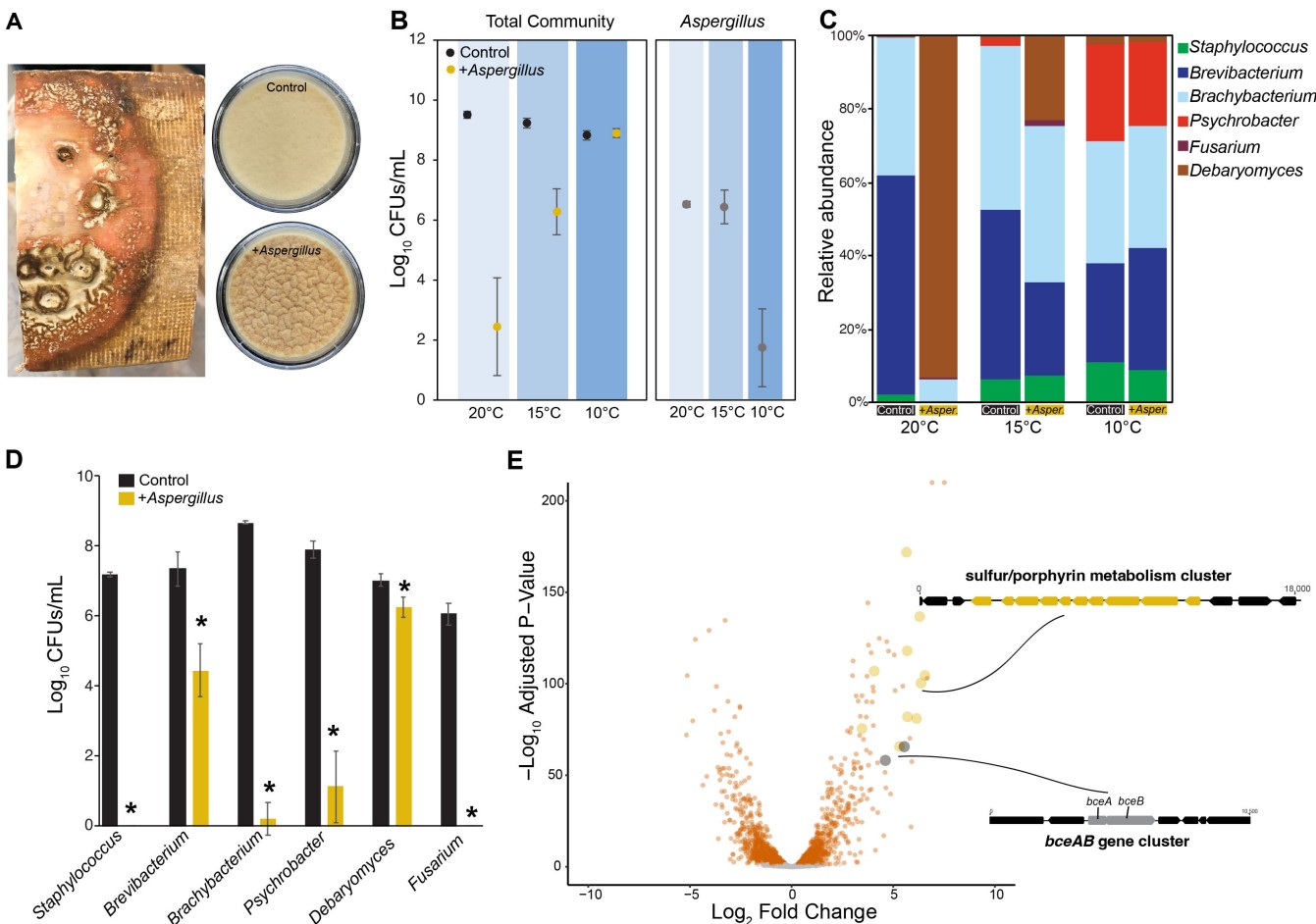

**FIG 1** *In situ* and *in vitro* impacts of *A. westerdijkiae* on a washed-rind microbial community. (A) On the surface of aging washed-rind cheeses (photo on left), *A. westerdijkiae* causes the typical orange rind to be discolored and sometimes develop wrinkles. In simulated cheese rinds in the lab, the same wrinkle phenotype appears after 21 days of rind development (photos of plates on right). (B) Total CFUs from experimental cheese-rind communities aged at different temperatures (20°C, 15°C, and 10°C). The panel on the left shows only CFUs for the experimental rind community members without *A. westerdijkiae* counts. The panel on the right shows counts for *A. westerdijkiae,* with a significant reduction of CFUs at 10°C. Data are from three experimental replicates with four technical replicates in each experiment. Points indicate mean values, and error bars are one standard deviation of the mean. (C) The presence of *A. westerdijkiae* causes shifts in community composition, but only at 20°C and 15°C. The plotted relative abundance data is the mean from three experimental replicates with four technical replicates in each experiment. (D) Pairwise interactions between typical cheese-rind community members and *A. westerdijkiae*. Data are from five technical replicates. Each bar represents average CFUs/mL and error bars are one standard deviation of the mean. Asterisks indicate significant differences in CFU concentrations based on two-sample tests ($P < 0.05$). (E) Transcriptional response of *S. equorum* strain BC9 to the presence of *A. westerdijkiae* after 96 h of co-culture on cheese curd. Each dot in the volcano plot represents a gene in the *S. equorum* genome, and those highlighted in orange or other colors are significantly up- or down-regulated. Two gene clusters that are highly up-regulated in the presence of *A. westerdijkiae* are shown.

well-studied focal bacterium, *Staphylococcus equorum*, to understand bacterial responses to *A. westerdijkiae*. This is a very well-studied food-associated *Staphylococcus* species with considerable past transcriptome studies, allowing us to compare transcriptional response to *A. westerdijkiae* to other interactions (25, 26). We then focused our assessment of the dynamics of metabolite production with *S. equorum* and another rind bacterium, *Brachybacterium alimentarium*. Specifically, we hypothesized that *A. westerdijkiae* alters its metabolic output in response to bacterial competitors, via chemical defenses. Furthermore, we explored whether the identified fungal metabolites have species-specific effects on four microbial community members. By elucidating the role of fungal secondary metabolites in shaping microbial communities, this work contributes to broader insights into microbial competition and the dynamics of microbial interactions in fermented food ecosystems.

## RESULTS

### *Aspergillus westerdijkiae* inhibits growth and alters gene expression of cheese-rind bacteria and communities

We first evaluated the impact of *A. westerdijkiae* on a model washed-rind cheese community by comparing a control community to one supplemented with this spoilage mold. The experimental communities contained a full suite of bacteria and fungi that represent some of the dominant microbes found on washed-rind cheeses: the bacteria *S. equorum* strain BC9 (Firmicutes), *Brevibacterium aurantiacum* strain BW86/JB5 (Actinobacteria), *B. alimentarium* strain BW87/JB7 (Actinobacteria), and *Psychrobacter* sp. strain BW96/JB193 (Proteobacteria); the yeast *Debaryomyces* sp. strain 135B; and the mold *Fusarium domesticum* strain 554A. In this initial set of community experiments, we included all typical community members, including both bacteria and fungi, to understand the full range of interactions between *A. westerdijkiae* and these microbes. In the mechanistic experiments that follow below (MALDI-MSI, liquid chromatography-tandem mass spectrometry [LC-MS/MS], penicillic acid [PA] assays, etc.), we only focused on bacteria because of their dominance in washed-rind cheese communities and because of their strong responses to *A. westerdijkiae*. Our mechanistic experiments below often focused on just *S. equorum* and *B. alimentarium,* for reasons explained in each section below. All experiments used a single *A. westerdijkiae* strain (Aw2019).

Both the control communities and those with *A. westerdijkiae* added were inoculated onto cheese curd agar (CCA) and incubated for 21 days at three different temperatures: 10°C, 15°C, and 20°C. While traditional cheese caves are typically maintained at 10°C–14°C to support proper rind development, temperature fluctuations, due to seasonal heat or equipment failure, can lead to elevated cave temperatures. We used a range of temperatures to understand if abnormal temperature conditions could promote the growth of *A. westerdijkiae* and facilitate its negative impacts on rind development.

Total growth of beneficial rind bacteria and fungi (excluding *Aspergillus*) was significantly lower in the 20°C and 15°C conditions, but not at the coldest temperature of 10°C (Fig. 1B), with very large reductions in CFUs at 20°C (two-way ANOVA temperature F-value = 91.3, $P < 0.001$; *A. westerdijkiae* F-value = 355.6, $P < 0.001$; temperature × *A. westerdijkiae* interaction F-value = 138.2, $P < 0.001$). There was no significant difference between the Control and +*A. westerdijkiae* in total CFUs at 10°C (Tukey's *post hoc P* = 0.999). When considering community composition, based on a two-way PERMANOVA, temperature (PERMANOVA $F = 20.47$, $P < 0.001$), *A. westerdijkiae* ($F = 36.55$, $P < 0.001$), and the interaction between the two factors ($F = 19.17$, $P < 0.001$) had significant effects on community structure (Fig. 1C). *Post hoc* tests showed that effects of *A. westerdijkiae* on community structure were only observed at the higher temperatures of 20°C and 15°C, and not at 10°C. These experimental cheese-rind community data demonstrate that the presence of the spoilage mold *A. westerdijkiae* can strongly inhibit desirable cheese-rind bacteria, but these effects are only observed at elevated temperatures.

To understand how each individual member of the experimental community interacts with *A. westerdijkiae*, we conducted pairwise co-culture assays with the same species used in the community experiments. Just as we observed with whole-community experiments above, CFU counts revealed that *A. westerdijkiae* significantly suppressed the growth of most microbial strains compared to their respective monocultures (Fig. 1D). All bacterial species, but especially *Staphylococcus equorum* and *Brachybacterium alimentarium,* exhibited marked growth inhibition in the presence of the fungus, suggesting antagonistic interactions. The filamentous fungus *Fusarium domesticum* was also highly inhibited, with no growth detected when *A. westerdijkiae* was present. The yeast *Debaryomyces hansenii* was the least inhibited of all microbes tested in co-culture assays, suggesting an ability to tolerate the inhibitory effects of *A. westerdijkiae* or that rapid growth of the yeast before *A. westerdijkiae* can grow allows it to escape the antagonistic effects of this spoilage fungus.

As an initial step toward understanding the inhibitory mechanisms of *A. westerdijkiae*, we carried out transcriptomic profiling of *S. equorum* under monoculture and co-culture conditions with this spoilage mold. We selected *S. equorum* because it was strongly inhibited by *A. westerdijkiae* and because we have extensive transcriptomic data from its interactions with typical, non-inhibitory rind fungi, allowing for meaningful comparisons (25, 26). We predicted that genes upregulated in the presence of *A. westerdijkiae* would indicate how *S. equorum* responds to this mold, and that genes uniquely differentially expressed, compared to past co-cultures with beneficial fungi like *Penicillium* spp., would highlight bacterial responses specific to inhibitory fungal interactions.

Differential gene expression analysis after 4 days of co-culture revealed a strong transcriptomic response of the bacterium to the presence of *A. westerdijkiae,* with 417 upregulated and 488 genes downregulated in *S. equorum* when grown with *A. westerdijkiae* (Fig. 1E). This differential expression of 32% of predicted genes in the genome is similar to previous transcriptomic responses of *S. equorum* to other cheese-rind fungi (mostly benign *Penicillium* species) where 31 to 46% of *S. equorum* genes responded to the presence of fungal neighbors (25). Pathway enrichment analysis identified strong transcriptional responses in *S. equorum* that were similar to what we have observed in previous co-culture RNA-seq studies, including upregulation of various nutrient acquisition pathways (sulfur metabolism, porphyrin production, thiamine metabolism, and biosynthesis of amino acids; Table S1). One gene cluster that was strongly upregulated and is an unusual transcriptomic response in *S. equorum* is the predicted *bceAB* cluster. BceAB transporters provide various Firmicutes with resistance to antimicrobial compounds that target cell wall synthesis and also play roles in bacterial responses to a variety of stresses (27, 28). The only other time we have observed upregulation of this gene cluster in *S. equorum* is when it was grown with the highly inhibitory fungus *Penicillium chrysogenum*, which produces penicillin and other antimicrobial compounds (25). We are not sure what role this putative *bceAB* cluster and its protein products may play in bacterial-fungal interactions, but it suggests that when bacteria are co-cultured with inhibitory fungi like *A. westerdijkiae,* they may use systems like BceAB transporters to deal with antimicrobial compounds produced by the fungi, which would support our initial hypothesis.

## Predicted metabolites of *A. westerdijkiae* and their spatially targeted production during interactions with bacteria

To investigate the potential antimicrobial compounds underlying the observed inhibition and transcriptional responses, we analyzed the *A. westerdijkiae* genome using antiSMASH to predict its BGCs (Table S2) (24). This analysis and Clinker-based comparison of these fungal BGCs reveal high synteny and sequence identity of at least 50% with known clusters for OTA, ochrindole A, and notoamide A, especially among core biosynthetic genes (Fig. S1). These findings suggest that *A. westerdijkiae* possesses the biosynthetic machinery for producing these bioactive compounds and structural analogs that share similar core scaffolds.

Given this high biosynthetic potential, we expanded our metabolite search beyond exact mass measurements by examining the MALDI-MSI data for ion signals corresponding to predicted adducts of both the BGC-associated compounds and their known analogs described in the literature. We performed MALDI-MSI on co-cultures of *A. westerdijkiae* and *B. alimentarium* according to the experimental setup shown in Fig. 2A. Phenotypically, co-culture zones between *A. westerdijkiae* and *B. alimentarium* defined two key regions of interest: the fungal-bacterial interaction (FBI) zone, representing the direct interface between the two species, and the bacterial-bacterial interaction (BBI) zone, an area encompassing a gradient of bacterial crosstalk without fungal contact. This design enabled us to compare the spatial distribution of fungal metabolites and assess how far these signals diffuse beyond the interaction interface. It is important to note that we did not use *S. equorum* as a responding bacterium in these MALDI-MSI interaction experiments, as we did with the RNA-sequencing experiments above, because *B.*

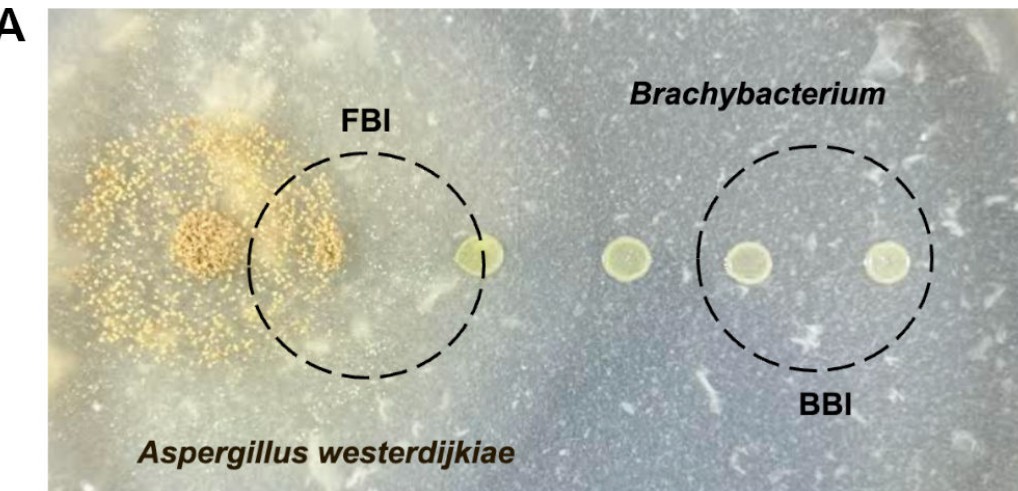

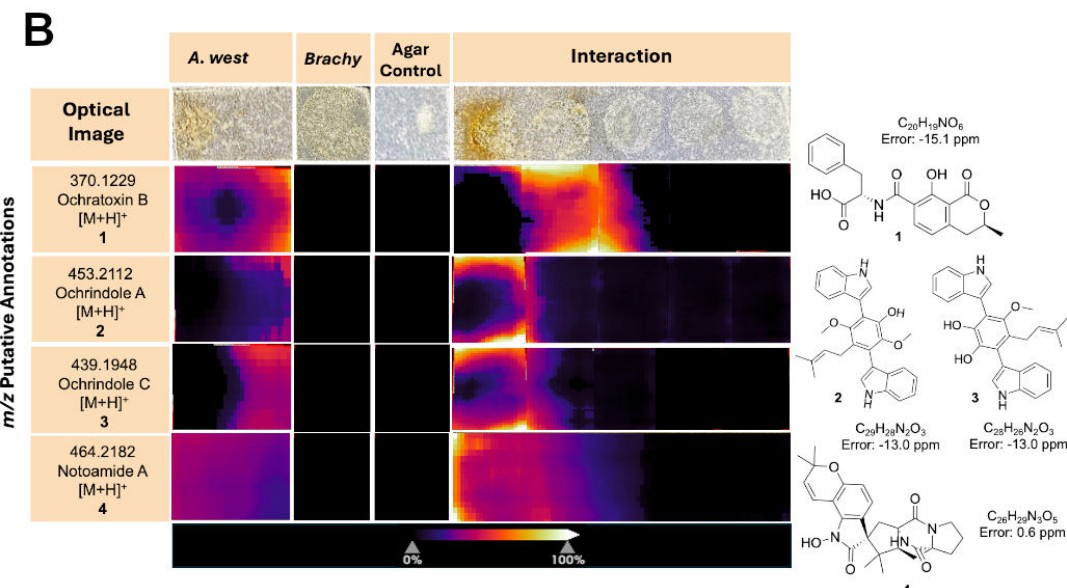

**FIG 2** Co-culture model to visualize spatially localized bioactive molecules detected via MALDI-MSI. (A) Representative images of the co-culture interaction model between the bacterium *B. alimentarium* (indicated as *Brachybacterium* or *Brachy* in the figure) and the fungus *A. westerdijkiae* on cheese agar (day 5) employed for the acquisition of MALDI-MSI and LC-MS/MS data. Dashed circles indicate FBI and BBI zones evaluated with the analytical techniques. (B) Optical images (top row) and MALDI imaging mass spectrometry heatmaps showing spatial localization of putatively annotated fungal molecules during interaction between *A. westerdijkiae* and *B. alimentarium*. Compounds include ochratoxin B (1), ochrindole A (2), ochrindole C (3), and notoamide A (4). The abundance of *m/z* signals is shown as heat maps, normalized to total ion count intensities across the following conditions: *A. westerdijkiae* monoculture, *B. alimentarium* monoculture, agar control, and the co-culture interaction. Chemical structures and exact mass errors are shown on the right; ppm errors are derived from MSI data.

*alimentarium* grew more consistently across the interaction zones with *A. westerdijkiae* in the MALDI-MSI setup and was less inhibited than *S. equorum*.

While we did not detect OTA in our spatial analysis, we observed clear signals for ochratoxin B (OTB) (1), suggesting selective or condition-dependent production of specific analogs. Similarly, we detected ochrindole A (2) as well as its structural analog ochrindole C (3), along with notoamide A (4). These detected metabolites were found to localize predominantly in increased intensities at the FBI zone compared to the controls (Fig. 2B). This spatially resolved pattern suggests that *A. westerdijkiae* directs the production of specific secondary metabolites, including structural analogs, toward its bacterial competitor.

To enhance metabolite annotation and uncover additional fungal compounds, we analyzed extracts from these monocultures and co-cultures using LC-MS/MS, followed by classical molecular networking through the GNPS platform (23, 29). The molecular network revealed families of structurally related fungal metabolites, including PA (6), circumdatins A (7), B (8), and F (9) that were enriched during fungal interaction with *B. alimentarium* (Fig. 3A; Fig. S2 to S4). Notably, the spatial distribution of these compounds, as observed by MALDI-MSI, aligned with their enrichment in the FBI zone (9 was not detected in MSI data) (Fig. 3B), further supporting their role in interspecies interactions. This integrative analysis demonstrates that *A. westerdijkiae* not only possesses the genetic machinery to produce antimicrobial metabolites, but also deploys them in a spatially targeted manner during interactions with bacteria.

## LC-MS confirms production of mycotoxins OTB and PA

Among the metabolites annotated in our interaction assays, compounds 1 and 6 emerged as key candidates for validation based on two criteria: (i) their well-documented toxicity and relevance to food safety, and (ii) their prominent spatial localization in MALDI-MSI experiments, which suggested active secretion by *A. westerdijkiae* during microbial interaction. While other compounds were detected, these two metabolites stood out as representative mycotoxins with potential ecological and industrial impact. To confirm their identity, we performed LC-MS analysis on extracts obtained from *A. westerdijkiae* and *B. alimentarium* co-cultures, grown under the same spatial interaction setup used for MALDI-MSI. Co-injection with commercial standards showed co-elution of peaks corresponding to 1 and 6 (Fig. 4A). The exact masses of the standards matched the expected protonated molecules 1 (Calc. 370.1285, obsv. 370.1286, 0.27 ppm) and 6 (Calc. 171.0652, obsv. 171.0656, 2.3 ppm) in the extracts (Fig. 4B). In addition, chromatograms of crude extracts in the absence of standards revealed higher relative ion intensities for 1 and 6 in co-culture conditions (light green chromatograms) compared to fungal monocultures (light red chromatograms), suggesting that bacterial presence increases the production of these mycotoxins. These results confirm that *A. westerdijkiae* produces both mycotoxins under the conditions tested, reinforcing their proposed role in chemically mediated microbial interference within cheese-rind communities.

## Mycotoxin production varies with bacterial partner and interaction type

Next, we investigated whether the production of key fungal metabolites was influenced by the identity of the bacterial interaction partner. Extracted ion chromatograms (EICs) revealed that both compounds 1 and 6 were markedly upregulated in co-cultures with *B. alimentarium* compared to *A. westerdijkiae* monocultures (Fig. 5A). Notably, 6 was also detected, albeit at lower relative intensity, in the BBI, suggesting that this compound may diffuse beyond the immediate FBI interface. This observation is consistent with the MSI data showing a broader spatial distribution of 6 across the agar surface. In contrast, co-cultures with *S. equorum* did not exhibit a similar upregulation of either metabolite. Both 1 and 6 remained at levels comparable to those observed in monocultures or were undetectable in the bacterial self-interaction regions (Fig. 5B). These results indicate a species-specific fungal metabolic response, where *A. westerdijkiae* secretes higher levels of mycotoxins in the presence of *B. alimentarium* but not *S. equorum*, highlighting the context-dependent nature of fungal chemical defense. This observation is further supported by MALDI-MSI analysis, which shows comparable signal intensities for compounds 1 and 6 between the *A. westerdijkiae–S. equorum* co-culture and the respective monocultures (Fig. S5).

Given the higher relative ion abundance and broader spatial distribution of 6 observed in *Brachybacterium* co-cultures, we carried out targeted quantification of this mycotoxin across different interaction conditions. Using LC-MS, we first extracted the peak areas for PA from the EICs in each sample and then converted these values to absolute concentrations using a standard calibration curve generated with purified PA

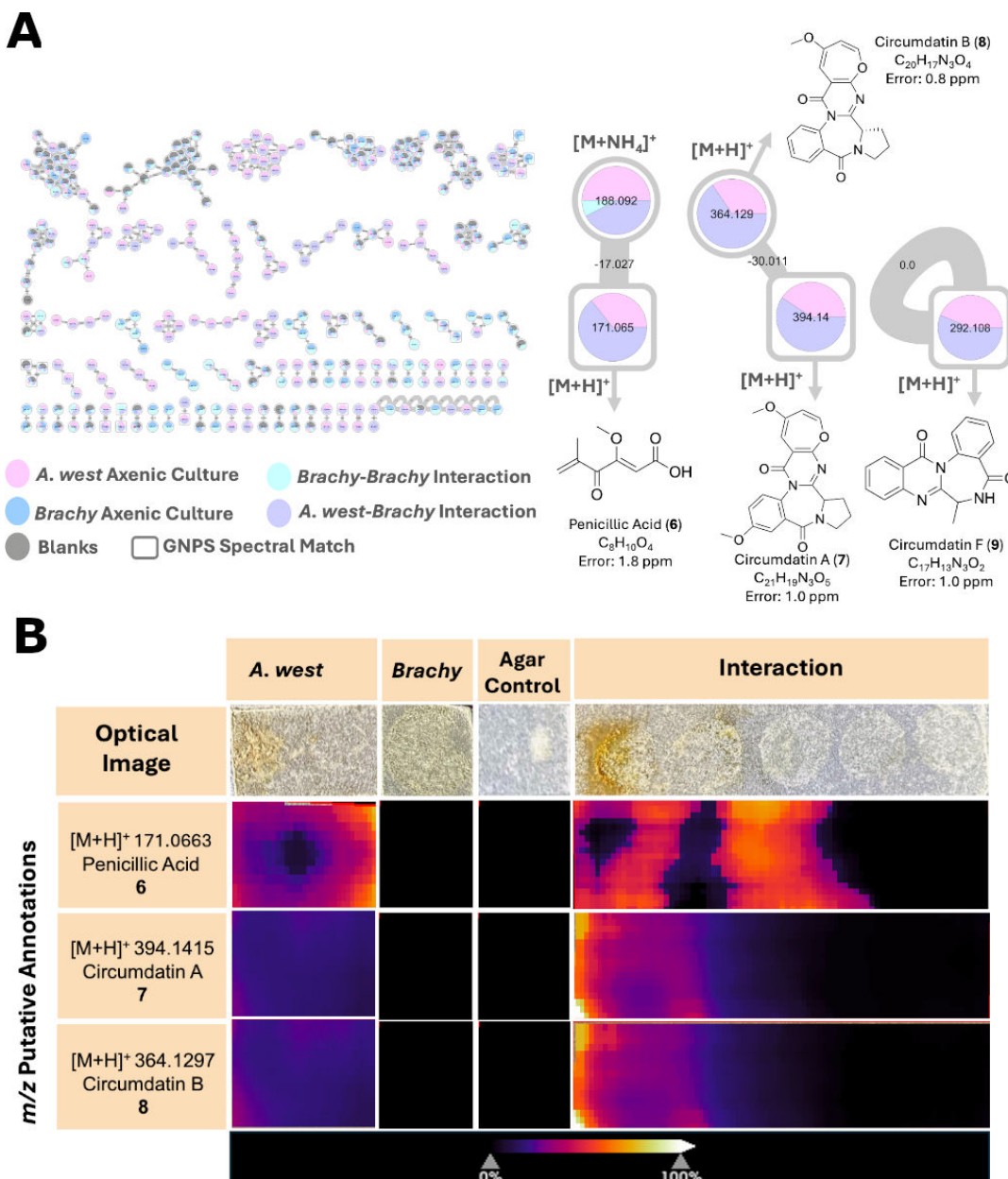

FIG 3 Integrating molecular networking and MALDI-MSI to enhance annotation of fungal metabolites during microbial interactions. (A) Overview of classical molecular networks generated from LC-MS/MS data across monocultures and interaction conditions of *A. westerdijkiae* (abbreviated as "*A. west*") and *B. alimentarium* (abbreviated as "*Brachy*") (left). Node colors represent sample origin: *A. westerdijkiae* monoculture (pink), *B. alimentarium* monoculture (blue), *A. westerdijkiae-B. alimentarium* interaction (purple), and *B. alimentarium-B. alimentarium* interaction (cyan). Gray nodes are features found in blanks. Square-shaped nodes indicate GNPS library spectral matches. Molecular families containing annotated compounds include penicillic acid (6), circumdatin A (7), and circumdatin B (8) are shown (right). (B) Optical images and MALDI imaging mass spectrometry heat maps show spatial localization of key metabolites from panel A. Penicillic acid and the circumdatins were predominantly detected in the *A. westerdijkiae-B. alimentarium* interaction zone, supporting their production during interspecies interaction. The abundance of *m/z* signals is shown as heat maps, normalized to total ion count intensities across the following conditions: *A. westerdijkiae* monoculture, *B. alimentarium* monoculture, agar control, and the co-culture interaction. The mass accuracy error is derived from LC-MS data and provided for each putative annotation.

(Fig. S6). Quantification was carried out for *A. westerdijkiae* monocultures, co-cultures with *B. alimentarium* and *S. equorum*, and the corresponding BBI zones.

As shown in Fig. 6A, raw peak areas were markedly higher in *A. westerdijkiae–B. alimentarium* co-cultures relative to the fungal monoculture and the *S. equorum* interaction. When these peak areas were converted to concentrations (Fig. 6B),

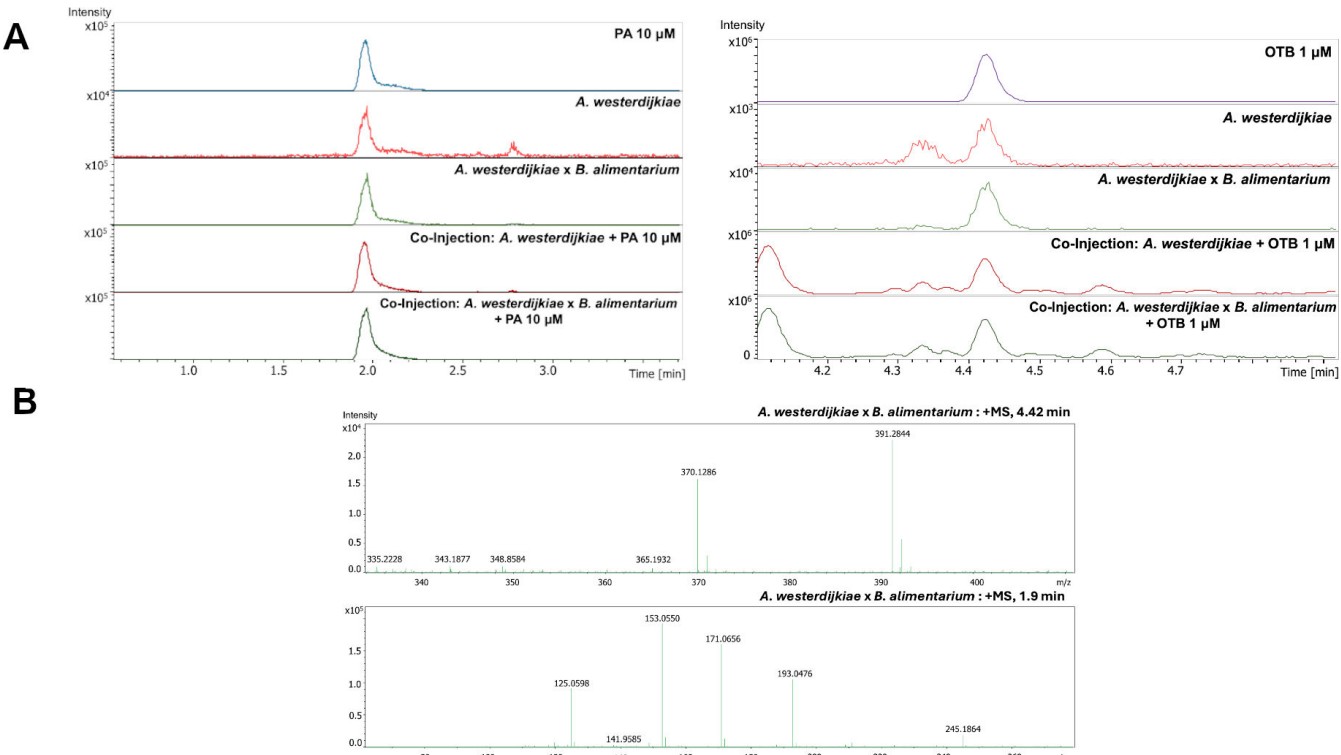

**FIG 4** LC-MS validation of mycotoxin production by *Aspergillus westerdijkiae* using commercial standards. (A) Extracted ion chromatograms (EICs) of OTB (right) and penicillic acid (PA, left) from *A. westerdijkiae* monoculture, co-culture with *Brachybacterium alimentarium*, and co-injections with analytical standards. Co-elution between standard and sample peaks confirms the presence of OTB (1 µM, retention time 4.42 min) and PA (10 µM, retention time 1.9 min) in microbial extracts. (B) MS spectra of OTB (top) and PA (bottom) in the FBI extracts.

co-cultures with *B. alimentarium* consistently showed the highest levels of PA (approximately 9.4 µM). In contrast, *A. westerdijkiae* monocultures and *S. equorum* co-cultures produced substantially lower concentrations (approximately 3.7 µM and 1.9 µM, respectively). We also detected PA in the BBI zone of the *Brachybacterium* plates, though at lower concentrations than in the direct FBI. This pattern suggests diffusion of the metabolite away from the interaction zone, diminishing with increasing distance from the interaction interface, which is aligned with the MSI and ESI data (Fig.3 and 5, respectively) shown before.

This general pattern supports the hypothesis that bacterial identity modulates the extent of mycotoxin production by *A. westerdijkiae*, potentially as a targeted response to specific microbial species. Consistent with this partner-specific response, bromocresol purple pH-indicator assays (Fig. S7; Table S3) showed that *A. westerdijkiae* acidifies the medium in both monoculture and co-culture, indicating that the elevated penicillic acid levels observed in *B. alimentarium* interactions cannot be attributed solely to bulk pH effects.

Since compounds 1 and 6 appeared to be more prominently produced during interaction with *B. alimentarium*, we next asked whether *A. westerdijkiae* deploys alternative chemical strategies when interacting with *S. equorum*.

In this case, we performed multivariate analysis that included PCA and PLS-DA on the LC-MS/MS data sets from both co-cultures and the corresponding controls. The resulting scores plots revealed clear separation between the *A. westerdijkiae × B. alimentarium* and *A. westerdijkiae × S. equorum* interactions, indicating that the fungus adopts distinct chemical phenotypes in each context (Fig. S8A through D). Examination of the loadings showed that compound 6 strongly influenced the position of the *B. alimentarium* co-cultures, consistent with its prominent upregulation in this interaction. In contrast, the

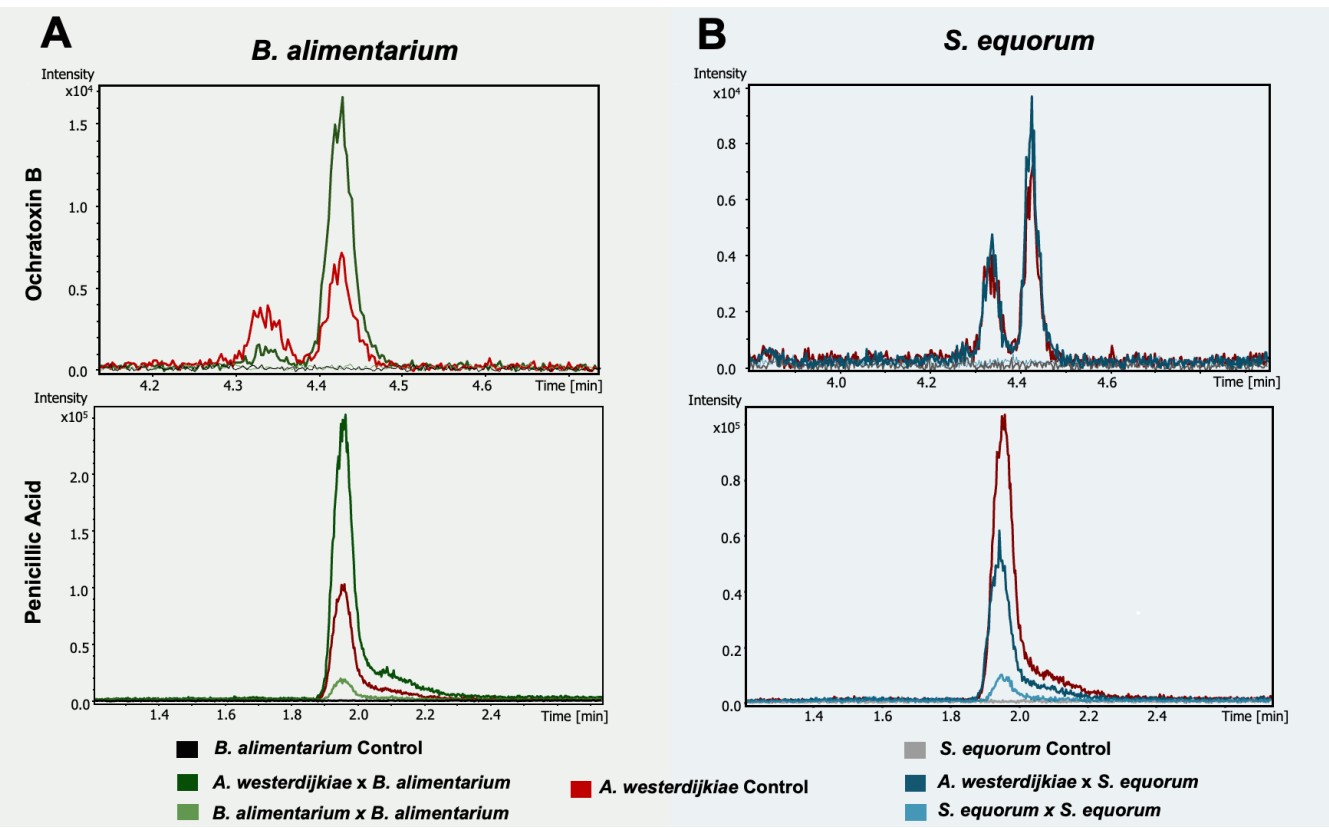

**FIG 5** PA and OTB fungal responses across different microbial interactions. (A) EICs showing peak areas of OTB (top) and PA (bottom) during interactions with *Brachybacterium alimentarium*. Co-culture of *A. westerdijkiae* with *Brachybacterium* (dark green) leads to increased production of both metabolites compared to monocultures (black and red traces). Additional signal is observed in *Brachybacterium-Brachybacterium* self-interaction (light green), suggesting metabolite diffusion. (B) Comparable EICs for *S. equorum* interactions reveal that both OTB and PA are also produced during fungal interaction with *Staphylococcus* (dark blue), comparable to those observed in monocultures.

separation of the *S. equorum* co-cultures was driven by a different set of metabolites, notably notoamides 4, 10, and 11, which were selectively enriched in this interaction but minimally detected with *B. alimentarium*. Together, these multivariate patterns demonstrate that *A. westerdijkiae* deploys a partner-specific chemical arsenal, adjusting its biosynthetic output depending on the identity of the competing bacterium.

## PA exhibits antimicrobial activity against cheese-rind bacteria

To confirm the impact of PA on the growth of individual cheese-rind bacteria, we supplemented CCA with different concentrations of this purified mycotoxin (10, 100, and 1,000 µM) and monitored them over 10 days (Fig. 7). The results revealed distinct, species-specific responses. In agreement with the chemical data above, *Brachybacterium* showed a clear dose-dependent inhibition, with growth increasingly suppressed as PA levels rose. In contrast, *Staphylococcus* appeared largely unaffected, exhibiting minor growth reductions, and mostly impacted only at the highest concentration. *Psychrobacter* proved to be especially sensitive, with PA strongly inhibiting its growth at all concentrations tested. Interestingly, *Brevibacterium* responded in the opposite manner at lower doses, displaying enhanced growth at 10 and 100 µM, though this stimulatory effect was lost at 1,000 µM, where inhibition was again observed.

## DISCUSSION

In this study, we integrated community and pairwise multiomics to investigate the chemical signaling of the spoilage fungus *A. westerdijkiae* against bacteria from

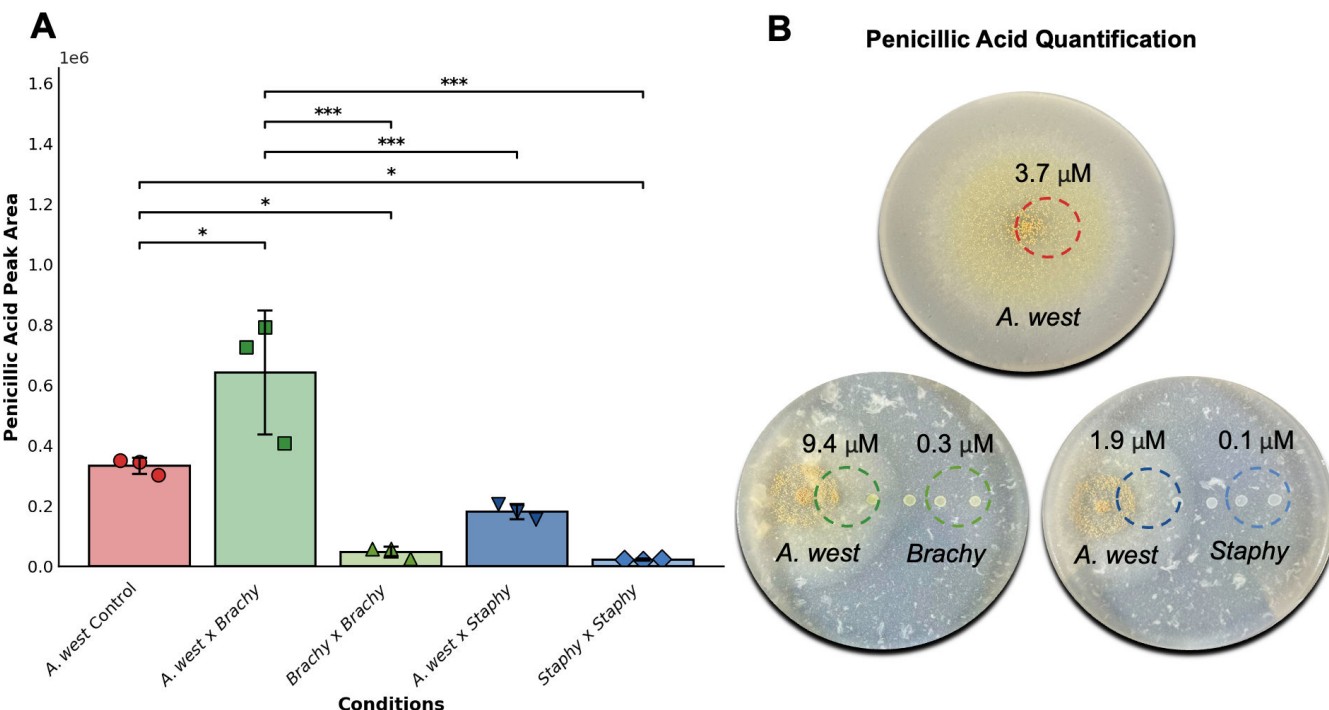

**FIG 6** Comparison of PA peak area and quantified concentrations across different culture conditions. (A) Peak areas obtained from EICs for PA. These values represent the raw LC-MS peak areas used as input for quantification in different experimental conditions: monoculture of *A. westerdijkiae* (abbreviated "*A.west*," red), the interaction interface between *A. westerdijkiae* and *B. alimentarium* (abbreviated "*Brachy*," dark green), the interaction interface between *A. westerdijkiae* and *S. equorum* (abbreviated "*Staphy*," dark blue), and the bacterial side of the interaction between *B. alimentarium* × *B. alimentarium* (light green) and *S. equorum* × *S. equorum* (light blue). Individual data points represent biological replicates ($n = 3$). (B) Absolute concentrations of PA (µM) represented in the defined sampling zones on cheese agar plates, calculated by correlating the peak areas from panel (A) with the 8-point calibration curve generated using analytical standard (0.1 µM–25 µM; Fig. S6). Values represent PA levels in the fungal control, FBI, and BBI zones (represented as dashed circles) across three plate conditions. Statistical analysis was performed using one-way ANOVA followed by Tukey's Honest Significant Difference test for multiple comparisons. Statistically significant differences are indicated by asterisks (*, **, ***), where $P < 0.05$ (*), $P < 0.01$ (**), and $P < 0.001$ (***).

washed-rind cheeses. Our initial interaction assays showed that *A. westerdijkiae* strongly inhibits the growth of common cheese rind-associated bacteria, including *B. alimentarium* and *S. equorum*, and induces transcriptional changes in *S. equorum*. These findings prompted the development of a spatial model system using CCA which allowed us to analyze microbial metabolites in a spatially structured environment (8). Our results demonstrate that *A. westerdijkiae* exhibits species-specific secondary metabolite production, with selective deployment of mycotoxins and other bioactive compounds in response to different bacterial competitors. These findings highlight how spoilage fungi alter microbial community dynamics through context-dependent chemical strategies.

*A. westerdijkiae* produces distinct metabolites that localize to the FBI, suggesting spatially directed chemical interference. Compounds such as OTB, ochrindoles A and C, notoamide A, PA, and circumdatins A and B were specifically enriched at the site of interaction with *B. alimentarium*, indicating that metabolite production is not uniform but rather spatially regulated in response to competitor presence. The spatial distribution of these metabolites also aligns with their known biological activities, reinforcing their functional relevance in microbial competition. OTB, a non-chlorinated analog of OTA, exhibits nephrotoxic and immunosuppressive properties and can inhibit bacterial growth, although it is less potent than OTA (30, 31). Ochrindoles are bisindole alkaloids produced by *Aspergillus* species and reported to possess moderate insecticidal properties (32). Notoamides, another class of indole-derived fungal metabolites, have been associated with cytotoxic and signaling-disruptive effects (33, 34). PA is a low-molecular-weight mycotoxin produced by several *Penicillium* and *Aspergillus* species, with both cytotoxic and antimicrobial properties (35–37). Circumdatins, a group of benzodiazepine

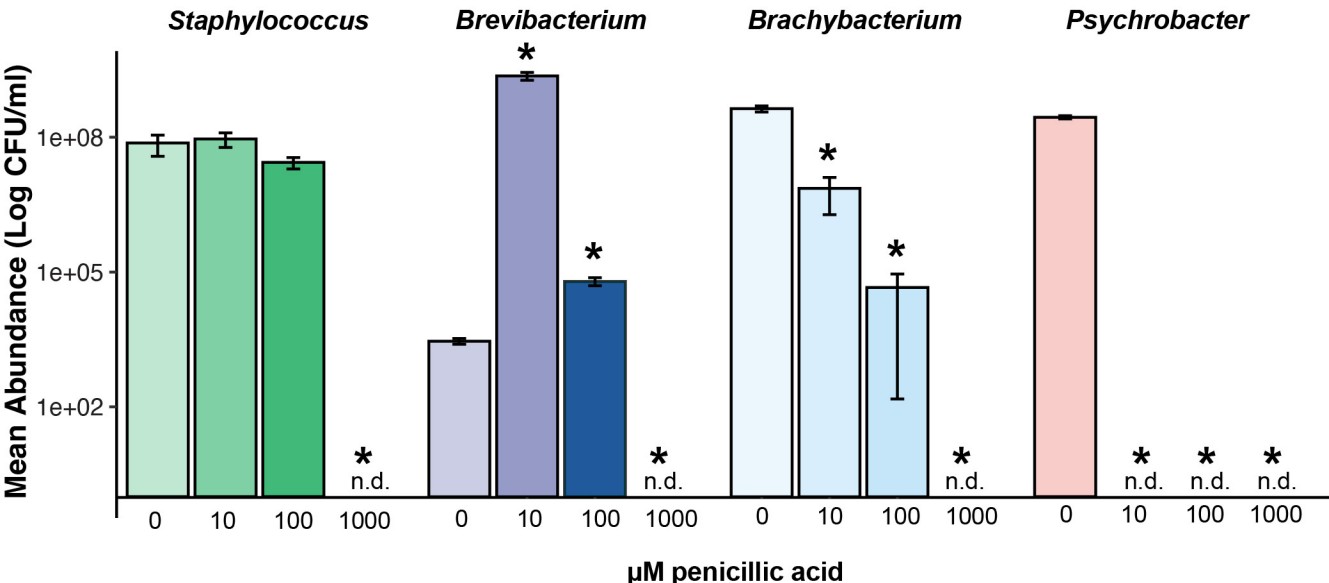

**FIG 7** Growth of cheese-rind bacteria on cheese curd with different levels of PA demonstrates antimicrobial activity of the pure compound. Bacteria (*Staphylococcus equorum* strain BC9, *Brachybacterium alimentarium* strain JB7, *Brevibacterium aurantiacum* strain JB5, and *Psychrobacter* sp. strain JB193) were grown on solid cheese curd for 10 days, and total growth was assessed across a range of PA concentrations. An asterisk over a bar in the 10, 100, or 1,000 µM PA concentrations indicates that growth was different from the no-pure-compound treatment based on an ANOVA with a Dunnett's test (*P* < 0.05). "n.d." = no CFUs detected.

derivatives, exhibit low antimicrobial, radical scavenging, and mitochondrial respiratory chain inhibitory activities (38–40).

Despite the presence of a BGC with high similarity to OTA, we consistently detected only OTB in our data sets. This suggests that the final chlorination step converting OTB to OTA may be environmentally regulated, potentially requiring specific abiotic or biotic conditions not replicated in our experimental setup (41). Context-sensitive activation of late-stage tailoring enzymes has been observed in other mycotoxin pathways and may involve regulatory cues tied to pH, oxidative stress, or microbial signaling (42). Alternatively, it is also possible that OTA is produced in low amounts relative to OTB and falls below the limit of detection of our analytical methods. In our experiments, even OTB was detected at relatively low concentrations based on its similar peak intensity evaluation with the standard (approximately 1 µM). Therefore, trace levels of OTA cannot be ruled out and may require more sensitive targeted methods, such as isotope dilution or immunoassay-based detection, for confident identification.

Interestingly, the spatial metabolomics data revealed that the mycotoxins, PA and OTB, were distributed over broader areas compared to other metabolites. This broader diffusion may reflect their higher potency and functional importance in fungal defense, enabling *A. westerdijkiae* to exert influence beyond the immediate interaction zone. It is worth noting that PA is often co-produced with ochratoxins by several *Aspergillus* species and has been shown to enhance the nephrotoxicity of OTA (43, 44). While it remains unknown whether a similar synergistic effect occurs between PA and OTB, their co-occurrence and spatial overlap in our system raise concerns about potential additive or synergistic toxic effects in food contexts. Moreover, drawing from recent findings by Vertot et al., *Penicillium hordei* was shown to secrete organic acids to acidify the environment and induce the precipitation of antifungal lipopeptides from *Bacillus subtilis* (45), it is worth considering whether PA might contribute to fungal defense through similar environmental modulation.

Our data reveal that *A. westerdijkiae* modulates its secondary metabolite profile depending on the identity of the interacting bacterial species. When co-cultured with *B. alimentarium*, the fungus produced significantly higher levels of PA and OTB. These

compounds were not similarly induced during interaction with *S. equorum*, pointing to a tailored response rather than a general stress reaction. The differential timing or growth rate of *S. equorum* relative to *B. alimentarium* may also influence the activation of these biosynthetic pathways, particularly for metabolites like PA, which may require sustained or specific cues to be expressed. Similar species-specific responses have been documented in other fungal systems, where interactions with different microbial partners or environmental contexts activate distinct BGCs, leading to unique metabolite profiles. For example, co-culture of *Aspergillus* with *Streptomyces* species triggered the production of novel secondary metabolites not observed in monoculture. Likewise, environmental signals such as microbial cross-talk, oxidative stress, or nutrient limitation have been shown to drive context-dependent mycotoxin biosynthesis (46–48). These findings support the idea that *A. westerdijkiae* tailors its chemical response to the microbial identity of its surroundings, deploying specific metabolites based on the ecological challenge it faces. Consistent with these species-specific chemical responses, the direct supplementation of purified PA revealed that individual cheese-rind bacteria vary markedly in their sensitivity to this metabolite. *Brachybacterium* and *Psychrobacter* were strongly inhibited even at low to intermediate PA concentrations, whereas *Staphylococcus* was relatively tolerant, showing substantial inhibition only at the highest dose. *Brevibacterium* displayed a stimulatory response at low PA levels, suggesting potential hormetic effects, but was inhibited at 1,000 µM. These growth phenotypes mirror the trends observed in our co-culture and spatial metabolomics analyses, strengthening the conclusion that *A. westerdijkiae* leverages PA as a selective, context-dependent chemical weapon within the cheese-rind microbiome.

More broadly, our findings support the idea that microbial communities are shaped not only by nutrient competition and environmental filtering but also by active, chemically mediated antagonism. Fungi like *A. westerdijkiae* use a modular and dynamic chemical arsenal that allows them to respond flexibly to ecological pressures, consistent with growing evidence from marine to plant-associated systems (49, 50).

## Conclusion

Although our model system captures key aspects of cheese-rind ecology, it is inherently simplified relative to the full complexity of *in situ* microbial communities. As microbial community assembly in natural environments involves additional abiotic gradients and microbial players, future work should evaluate whether these findings hold in multi-species systems or real cheese rinds. A particularly important next step would be to assess whether disrupting the biosynthesis of compounds like PA or OTB alters community composition. Together, our results demonstrate that *A. westerdijkiae* exhibits species-specific and spatially targeted chemical responses to bacterial competitors in the cheese-rind microbiome. We reveal how this invasive fungus modulates its secondary metabolism in response to microbial context, producing mycotoxins and structurally diverse metabolites in a selective, interface-directed manner. These findings provide new insights into fungal strategies of microbial interference in a fermented food model system.

## MATERIALS AND METHODS

### Microbial strains

Bacterial strains (*Staphylococcus equorum* str. BC9, *Brevibacterium aurantiacum* str. BW86/JB5, *Brachybacterium alimentarium* str. BW87/JB7, *Psychrobacter* sp. str. BW96/JB193) and fungal strains (*A. westerdijkiae* str. Aw2019, *Debaryomyces* sp. str. 135B, *Fusarium domesticum* str. 554A) were used in pairwise competition experiments. These microbes were all isolated from cheese rinds aged in various aging facilities across the United States and characterized using 16S rRNA (bacteria) or ITS (fungi) sequencing (4).

## Community assays

To determine how the presence of *A. westerdijkiae* affects cheese-rind microbial community assembly, we set up experimental rind communities on the surfaces of 60 mm Petri dishes containing 10 mL of CCA (51). Communities were inoculated with each of the bacteria and fungi listed above at a concentration of 100 CFUs/µL in a total volume of 35 µL that was spread across the surface of the CCA. Two treatments were used: one with all bacteria and fungi except *A. westerdijkiae* (Control) to represent a typical washed-rind cheese community, and one with *A. westerdijkiae* added to the bacterial and fungal community mix *(+A. westerdijkiae)* to represent a washed-rind cheese invaded by *A. westerdijkiae.* Communities were incubated in the dark at 20, 15, and 10°C to simulate a gradient of temperatures that can be experienced during cheese production and aging, with 15°C and 10°C being the upper and lower limits of temperatures typically used in cheese caves to age cheeses. Four technical replicates of each community and temperature treatment were conducted across three experimental replicates. After 21 days, community composition within each experimental unit was determined by using a cork borer (6 mm wide) to remove a plug of the cheese curd containing the surface rind community. Each cheese curd plug was placed in a 1.5 mL microcentrifuge tube with 500 µL of 15% glycerol in phosphate-buffered saline (PBS). The contents were homogenized using sterile micropestles and plated on plate count agar with milk and salt (PCAMS) as previously described (51). To determine significant shifts in microbial community composition based on temperature and the presence of *A. westerdijkiae,* we used a two-way PERMANOVA.

## Interspecies interaction assays

Microbial stocks (30% glycerol in 1× PBS) were thawed and diluted to 100 CFU/µL. Monoculture inocula were prepared by mixing each strain 1:1 with PBS. For pairwise co-cultures, bacterial strains were mixed 1:1 with *Aspergillus westerdijkiae* at the same concentration. A 10 µL inoculum was added to 1.5 mL microcentrifuge tubes containing 250 µL of CCA (25 g/L Bayley Hazen Blue cheese curd, 5 g/L xanthan gum, 30 g/L NaCl, 17 g/L agar, pH 7.2, adjusted with NaOH). Each of the 25 treatments (13 monocultures and 12 co-cultures) was set up in five biological replicates ($n = 5$).

Tubes were incubated in the dark at 20°C with lids open and sealed with sterile AeraSeal film (Millipore Sigma, #A9224) to allow gas exchange while minimizing contamination. Open tubes of deionized water were placed in racks to reduce media desiccation. Tube placement was randomized within racks to control for spatial variation in temperature and humidity.

After 7 days, 500 µL of 15% glycerol in PBS was added to each tube, and contents were homogenized using sterile pestles. Homogenates were stored at −80°C until colony enumeration. Monocultures were serially diluted and plated on PCAMS or brain heart infusion (BHI) agar. For co-cultures, serial dilutions were plated on PCAMS containing 50 mg/L chloramphenicol to inhibit bacterial growth (for *A. westerdijkiae* counts), and on PCAMS or BHI with 21.6 mg/L natamycin to inhibit fungal growth (for bacterial counts) (51). Due to overlapping antimicrobial susceptibilities, *A. westerdijkiae* was preferentially grown at 28°C, while competitor strains were incubated at 20°C to suppress fungal growth.

Colony counts were analyzed using JMP Pro v.17.2.0 (SAS Institute). One-way ANOVA followed by Dunnett's *post hoc* test was used to assess differences in *A. westerdijkiae* counts. Two-sample *t*-tests with unequal variances were used to compare monoculture and co-culture counts for each bacterial strain.

## RNA extraction, sequencing, and differential expression analysis

*S. equorum* (200 CFU/µL) and *A. westerdijkiae* (4 CFU/µL) were thawed and diluted in PBS. For monocultures, 100 µL of *S. equorum* was inoculated onto CCA with 100 µL of PBS. For co-cultures, 100 µL of *S. equorum* was mixed with 100 µL of *A. westerdijkiae* (1:50

fungal:bacterial CFU ratio). Plates were incubated at 20°C in the dark for 96 h ($n$ = 5 biological replicates per treatment).

Cells were harvested using a flame-sterilized razor blade and immediately submersed in 2 mL of RNAprotect Bacteria Reagent (Qiagen #76506), vortexed, and stored at −80°C for ≥24 h. After thawing, samples were centrifuged (4°C, 5,000 rpm, 10 min), and the supernatant was discarded. Cell pellets were resuspended in 200 µL of TE buffer (50 mM Tris-HCl, 50 mM EDTA, pH 8.0) containing 50 mg/mL lysozyme and 1.2% Triton X-100 and incubated at 37°C for 15 min.

Following cell wall disruption, cells were lysed in 500 µL 2× Buffer B (200 mM NaCl, 20 mM EDTA), 210 µL 20% SDS, and 500 µL phenol:chloroform:isoamyl alcohol (125:24:1, pH 4.5) with ~250 µL of mixed-diameter acid-washed glass beads. Samples were bead-beaten using a vortex adaptor (2 min, high speed), centrifuged (8,000 rpm, 3 min), and the aqueous phase (~500 µL) was extracted. RNA was precipitated with 50 µL of 3 M sodium acetate and 500 µL of cold isopropanol and incubated at −80°C overnight.

Pelleted RNA was recovered by centrifugation (4°C, 13,000 rpm, 10 min), washed with 70% ethanol, and centrifuged again (4°C, 13,000 rpm, 5 min). Pellets were air-dried and resuspended in 50 µL of nuclease-free water. RNA cleanup and DNase treatment were performed using the RNA Clean & Concentrator-5 Kit (Zymo Research #R1013). DNA-free status was verified via PCR amplification of the bacterial 16S region and gel electrophoresis.

Bacterial rRNA was removed using the NEBNext rRNA Depletion Kit (Bacteria) with RNA Sample Purification Beads (NEB #E7860). Fungal rRNA depletion was achieved using a custom oligonucleotide pool (2 µM final concentration per probe; Integrated DNA Technologies) targeting *A. westerdijkiae* rRNA, used in conjunction with the NEBNext RNA Depletion Core Reagent Set (NEB #E7865).

RNA libraries were constructed using the NEBNext Ultra II RNA Library Prep Kit for Illumina (NEB #E7770), following the manufacturer's protocol for intact RNA. Libraries were pooled and sequenced on an Illumina NovaSeq X Plus (1 × 150 bp, single-end) at the Tufts University Core Facility Genomics (Boston, MA, USA).

Demultiplexing, quality control, and adapter trimming were performed using bcl2fastq Conversion Software (version 2.20). Illumina reads were mapped to a previously assembled *S. equorum* str. BC9 genome using Geneious Prime mapper (version 2024.0.4) at medium-low sensitivity, with reads mapping to multiple best matches randomly assigned. Expression levels of each sample were calculated using the built-in Geneious Prime function, with ambiguously mapped reads counted as partial matches. Differential expression between treatments was calculated using the DESeq2 plugin (52), with replicates of each growth condition grouped ($n$ = 4). Genes were considered differentially expressed if *S. equorum* expression when grown in co-culture with *A. westerdijkiae* was greater than doubled (log2 ratio > 1) or less than halved (log2 ratio < −1) compared to growth in monoculture at a corrected *P*-value <0.05, adjusted for false discovery rate using the Benjamini-Hochberg procedure.

## antiSMASH BGC annotation

BGCs were annotated from a previously sequenced genome of *Aspergillus westerdijkiae* strain Aw2019 using antiSMASH v.7.0 with default settings (24), including cluster detection for polyketide synthases (PKS), nonribosomal peptide synthetases, hybrid clusters, and other secondary metabolite biosynthetic pathways. Selection criteria included a minimum similarity of ≥50% to a known BGC associated with a characterized compound. Regions located at sequence contig edges were excluded to avoid incomplete or ambiguous annotations. Only BGC annotations linked to the production of fungal-derived compounds were considered. The compounds associated with the annotated BGCs were searched in mass spectrometry data considering different adduct forms, including [M+H]$^+$, [M+H-H$_2$O]$^+$, and [M+Na]$^+$. In addition, for compounds with known analogs, mass spectrometry data were also examined for different adducts potentially corresponding to these analogs. This step aimed to correlate predicted BGC

products with detected metabolites, increasing confidence in the annotation. BGCs from *Aspergillus westerdijkiae*, predicted via antiSMASH, were compared to known reference BGCs for OTA, ochrindole A, and notoamide A using Clinker (53). Pairwise comparisons were performed using GenBank-formatted cluster files, enabling visualization of gene synteny and sequence identity. Figures highlight conserved core genes and structural rearrangements between homologous clusters.

## Microbial culture conditions for MALDI-MSI

*Aspergillus westerdijkiae* was grown on PCAMS (1 g/L whole-milk powder, 1 g/L dextrose, 2.5 g/L yeast extract, 5 g/L tryptone, 10 g/L NaCl, 15 g/L agar) plates at room temperature (RT), and spores were harvested after 7 days or upon visible sporulation. Spores harvested from *A. westerdijkiae* were suspended in 1× PBS and normalized to an $OD_{600}$ of 0.1. *Brachybacterium* and *Staphylococcus equorum* strains were separately grown on PCAMS agar plates at 25°C for 2 days. Single colonies were then transferred to 10 mL of BHI (Bacto) liquid medium and incubated overnight at 25°C with shaking at 225 rpm. Bacterial cultures were subsequently normalized to an $OD_{600}$ of 0.01. For fungal and bacterial positive controls, 2 µL of each normalized culture were individually spotted at the center of a 100 × 15 mm CCA plate. To prepare the FBIs, 2 µL of the normalized fungal culture were spotted at the center of the CCA plate, followed by four sequential 2 µL of normalized bacterial culture spots placed 1 cm apart. All five spots were applied on the same day, and plates were incubated at RT for 10 days. After incubation, the cultures were directed to MALDI-MSI and LC-MS analyses, with triplicate samples prepared for each type of analysis separately. For MSI experiments, thin agar plates (10 mL of CCA in a 90 mm Petri dish) were prepared to facilitate ionization and signal detection. For LC-MS experiments, thicker plates (20 mL of CCA) were used to enhance metabolite extraction.

## MALDI-MSI analysis of microbial cultures

After the incubation period, fungal spores were removed from the control and co-culture plates using a sterile cotton swab moistened with acetonitrile (ACN). Samples, including isolated bacteria, fungi, bacterium-fungus co-spots, and agar controls, were excised from the agar plates using a sterile razor blade and transferred to an MSP 96-target ground-steel plate (Bruker Daltonics). A 53 µm stainless steel sieve (Hogentogler Inc.) was used to coat the steel target plate and colonies with MALDI matrix. The matrix consisted of a 1:1 mixture of recrystallized δ-cyano-4-hydroxycinnamic acid and 2,5-dihydroxybenzoic acid (Sigma). The plate was then placed in an oven with slow rotation at 40°C for approximately 4 h or until the agar had fully desiccated (54). Once dried, excess matrix was removed from the target plate with a stream of air. Pictures of agar spots on the steel plate were taken both before and after matrix application to document sample distribution (optical images). The prepared target plate was inserted into a Bruker timsTOF fleX qTOF mass spectrometer (Bruker Daltonics), where the optical images were aligned with the steel plate inside the instrument to guide data acquisition in specific regions using the software applications timsControl v.4.1 and FlexImaging v.7.2. Data were collected in positive ion mode over a mass range of 50 Da–800 Da, with calibration performed using red phosphorus. MALDI-MSI images were acquired at 500 µm spatial resolution (laser size: 229 × 229 µm) using the M5 defocus laser setting. Each raster point was sampled with 200 laser shots at 1,000 Hz. All ion images were generated using SCiLS Lab software version 2024c Pro (Bruker Daltonics). They had hot spot removal applied and were normalized using a total ion count (TIC) normalization.

## Metabolite extraction and LC-MS/MS sample preparation

Microbial growth was excised from the agar medium using 50 mL Falcon tubes, targeting specific regions of interest: the interface of FBI, BBIs, the center of fungal or bacterial monocultures, and uninoculated agar controls. This was done by gently pressing the edge of the Falcon tube onto the agar surface, effectively isolating the desired microbial

sample within the tube. 10 mL of ACN was added to each tube and then sonicated for 60 min to facilitate metabolite extraction. The samples were centrifuged at 4,000 rpm and 4°C for 12 min. The resulting supernatant was carefully separated from the agar pieces and transferred to a new 15 mL Falcon tube. The samples were then subjected to a second centrifugation step under the same conditions to further clarify the extract. The ACN extract was then transferred into pre-weighed 20 mL scintillation vials and dried *in vacuo*. Once dried, the samples were resuspended in a 50:50 ACN:MilliQ water solution and diluted to a final concentration of 1 mg/mL. The resuspended samples were transferred to Eppendorf tubes and centrifuged at RT for 12 min at 10,000 rpm to remove any remaining particulates. The clarified supernatant was then transferred into LC vials and analyzed via LC-MS/MS.

## LC-MS/MS analysis

A $C_{18}$ Phenomenex Kinetex 100 Å 50 × 2.1 mm UPLC column with a particle size of 1.7 µm was used on a Bruker Elute UPLC (Bruker Daltonik, Billerica, MA). The column was equilibrated to 95% A ($H_2O$ + 0.1% formic acid) and subjected to a 10 min gradient from 5 to 100% B (ACN + 0.1% formic acid) with an injection volume of 8 µL and a flow rate of 0.5 mL/min at 40°C. Data were acquired in Compass Hystar 6.2 software. High-resolution LC-MS/MS detection was performed on a Bruker timsTOF fleX (Bruker Daltonik, Billerica, MA) in positive mode from $m/z$ 50–2,000 (Table 1). Nebulizer gas ($N_2$) was set to 2.8 bar, and dry gas was set to 10.0 L/min. Dry temperature was set to 230°C. The ESI conditions were set with the capillary voltage at 4.5 kV. LC-MS/MS data were collected using data-dependent acquisition mode at an MS spectra rate of 10 Hz and MS/MS spectra rate at 16 Hz with the top 5 precursors selected for fragmentation at the collision energies and isolation widths shown below. The software used was Bruker timsControl 6.0.

## LC-MS data processing

To create molecular networks in the GNPS environment, the tandem mass spectra data were converted into .mzXML files using MSConvert (55) and submitted to the GNPS platform(23) with the following parameters: all MS/MS peaks within ±17 Da of the precursor $m/z$ were removed. Repeated spectra were grouped using the MS-Cluster algorithm (Classic Molecular Networking) with a precursor ion mass tolerance and fragment ion tolerance of 0.02 Da to generate consensus spectra, each represented as a node. Only consensus spectra containing at least two nearly identical spectra were considered. A molecular network was then constructed using the representative consensus MS/MS spectra, with edges filtered to retain only those with a cosine similarity score above 0.7 and at least four matching fragment peaks. Edges were maintained in the network only if each node appeared within the 10 most similar nodes of the other. The network spectra were subsequently matched against GNPS spectral libraries, which were filtered in the same manner as the input data. Library matches were retained if they had a cosine score above 0.7 and at least four matching peaks. The resulting output was imported and visualized as molecular networks in Cytoscape 3.4.0 (56).

A feature list was created with the raw MS1 Bruker files, processed using T-ReX 3D in MetaboScape (Bruker Daltonik), and used to perform the multivariate analysis. The minimum number of features for extraction and result was 3/18, so a feature

**TABLE 1** Details of LC-MS/MS analysis

| Mass *(m/z)* | Width *(m/z)* | Collision energy (eV) |
|---|---|---|
| 50 | 2 | 20 |
| 1,000 | 4 | 20 |
| 500 | 6 | 20 |
| 1,300 | 8 | 30 |

must have been present in three files to appear in the feature list. The peak intensity threshold was 2,500, retention time spanning the gradient only. Ions used for extraction include $[M + H]^+$, $[M + Na]^+$, $[M + NH_4]^+$, and $[M + K]^+$. Modifications included a loss of water $[M - H_2O + H]^+$. Ion intensities were normalized using Total Ion Count (TIC) normalization to account for technical variation between runs. Following feature list generation, PCA and PLS-DA multivariate analysis were made and visualized in Python using the scikit-learn, matplotlib, and seaborn packages within a Jupyter Notebook, an open-source, web-based interactive computing environment.

For PCA, the first two principal components (PC1 and PC2) were retained for visualization. PCA loadings were extracted to identify *m/z* features with the highest contributions to group separation. These features were prioritized for further annotation and biological interpretation. For PLS-DA, supervised multivariate modeling was performed using the PLSRegression function from scikit-learn, incorporating two latent variables (PLS1 and PLS2). The response matrix (Y) was one-hot encoded to represent experimental group membership. Variable Importance in Projection (VIP) scores were computed to quantify the contribution of each *m/z* feature to group discrimination. Features with VIP scores > 1.0 were considered most influential.

## Penicillic acid quantification

PA quantification was performed using a calibration curve with standard solutions at concentrations of 0.1, 0.25, 0.5, 1, 2.5, 5, 10, and 25 µM. Each concentration point was analyzed in triplicate. The detection and quantification were conducted using LC-MS with the same method described above. The calibration curve was constructed by plotting the peak area against the corresponding concentrations, and linear regression was applied to determine the concentration of PA in the samples. The statistical analyses for PA quantification were performed in Python using Jupyter Notebook with the LC-MS peak area values detected in the samples. One-way ANOVA was conducted using the Statsmodels package to assess differences between conditions, followed by Tukey's *post hoc* test for multiple comparisons. Data visualization and significance annotations were generated using Matplotlib and Seaborn packages.

## Bromocresol purple pH-indicator assay

Bromocresol purple (0.02 g/L) was added to 2.5% CCA to visualize pH changes during FBIs. Calibration plates were prepared by adjusting aliquots of the indicator medium to pH 3, 5, 7, and 9 (*n* = 4 each), pouring into Petri dishes (10 mL/plate), and photographing them under standardized lighting. Blue and Red channel intensities were extracted from the center of each plate using ImageJ (57) to generate a color-pH calibration curve with the channels ratio.

Interaction plates and fungal controls were prepared using the same indicator medium and incubated for 7 days in five replicates. Images were analyzed in ImageJ by extracting Blue/Red ratios from two regions of interest: the fungal delimitation zone (ROI 1) and a distal interface region near bacterial inocula (ROI 2). Estimated pH values for each ROI were obtained by interpolating the calibration curve. Raw measurements and calculated pH values are shown in Table S3.

## Antimicrobial assays with standards

Microbial stocks of *Staphylococcus equorum* str. BC9, *Brevibacterium aurantiacum* str. BW86/JB5, *Brachybacterium alimentarium* str. BW87/JB7, *Psychrobacter* sp. str. BW96/JB193 were thawed and diluted to 20 CFU/µL. Ten microliters were inoculated (approximately 200 CFUs) in 1.5 mL–mL microcentrifuge tubes containing 150 µL of CCA with 10 µM, 100 µM, and 1,000 µM of PA. Samples were incubated at 24°C undisturbed in the dark for 10 days. On day 10, 500 µL of 15% glycerol was added to tubes, and samples were homogenized by pestling. Samples were diluted and plated on PCAMS to get cell

counts. Each of the 16 treatments (four monoculture strains at four PA concentrations) was set up in five biological replicates.

## ACKNOWLEDGMENTS

This work was supported in part by the National Institute of General Medical Sciences of the NIH award R21GM148870 (L.M.S.), California Institute for Regenerative Medicine (CIRM), award EDUC-12759, the UCSC Institute for the Biology of Stem Cells (IBSC) (C.O.G.), the National Science Foundation CAREER/IOS/BIO award 1,942,063 (B.E.W.), and the Northeast Dairy Business Innovation Center award 02200-DBIC-G4H-25-07 (B.E.W.).

## AUTHOR AFFILIATIONS

[1]Department of Chemistry and Biochemistry, University of California Santa Cruz, Santa Cruz, California, USA
[2]Department of Biology, Tufts University, Medford, Massachusetts, USA

## AUTHOR ORCIDs

Carlismari O. Grundmann ⓘ http://orcid.org/0000-0003-2799-3527
Benjamin E. Wolfe ⓘ http://orcid.org/0000-0002-0194-9336
Laura M. Sanchez ⓘ http://orcid.org/0000-0001-9223-7977

## FUNDING

| Funder | Grant(s) | Author(s) |
| --- | --- | --- |
| National Institute of General Medical Sciences | R21GM148870 | Laura M. Sanchez |
| California Institute for Regenerative Medicine | EDUC-12759 | Carlismari O. Grundmann |
| Division of Integrative Organismal Systems | 1942063 | Benjamin E. Wolfe |
| Northeast Dairy Business Innovation Center | 02200-DBIC-G4H-25-07 | Benjamin E. Wolfe |

## AUTHOR CONTRIBUTIONS

Carlismari O. Grundmann, Data curation, Formal analysis, Investigation, Methodology, Validation, Visualization, Writing – original draft, Writing – review and editing | Christopher J. Tomo, Data curation, Formal analysis, Methodology, Validation, Visualization, Writing – original draft, Writing – review and editing | Julia L. Hershelman, Formal analysis, Methodology, Writing – review and editing | Benjamin E. Wolfe, Conceptualization, Data curation, Formal analysis, Funding acquisition, Project administration, Supervision, Visualization, Writing – original draft, Writing – review and editing | Laura M. Sanchez, Conceptualization, Funding acquisition, Project administration, Supervision, Writing – original draft, Writing – review and editing

## DATA AVAILABILITY

*Staphylococcus equorum* RNA-seq reads are in NCBI with the Bioproject ID PRJNA1314083. Raw Illumina reads for the *Aspergillus westerdijkiae* genome have been deposited in the NCBI SRA as BioProject ID PRJNA1171465. The MSI data used to create the spatial distribution ion heatmaps, LC-MS/MS data used to create the molecular networks, and LC-MS data used to do the quantification experiments have been deposited in MassIVE under the study identifiers MSV000098830, MSV000097876, and MSV000098841.

## ADDITIONAL FILES

The following material is available online.

### Supplemental Material

**Supplemental figures (mSystems01305-25-s0001.pdf).** Figures S1 to S8 and legends for Tables S1 to S3.
**Table S1 (mSystems01305-25-s0002.xlsx).** Functional pathways enriched with differentially expressed genes of *S. equorum*.
**Table S2 (mSystems01305-25-s0003.xlsx).** Predicted BGCs identified in *A. westerdijkiae* genome sequence via antiSMASH.
**Table S3 (mSystems01305-25-s0004.xlsx).** Complete data set for bromocresol purple pH calibration curve and semi-quantitative analysis of pH gradients during *A. westerdijkiae* interactions.

### Open Peer Review

**PEER REVIEW HISTORY (review-history.pdf).** An accounting of the reviewer comments and feedback.

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
