## [Reviewer comments · mSystems]

Spatial Metabolomics Reveals the Role of Penicillic Acid in Cheese Rind Microbiome Disruption by a Spoilage Fungus

Carlismari Grundmann, Christopher Tomo, Julia Hershelman, Benjamin Wolfe, and Laura Sanchez

Corresponding Author(s): Laura Sanchez, University of California Santa Cruz

Review Timeline:

Submission Date:	September 7, 2025
Editorial Decision:	October 6, 2025
Revision Received:	November 24, 2025
Accepted:	December 12, 2025

Editor: Trent Northen

Reviewer(s): Disclosure of reviewer identity is with reference to reviewer comments included in decision letter(s). The following individuals involved in review of your submission have agreed to reveal their identity: Scott A. Jarmusch (Reviewer #1)

Transaction Report:

DOI: <https://doi.org/10.1128/msystems.01305-25>

Re: mSystems01305-25 (Spatial Metabolomics Reveals the Role of Penicillic Acid in Cheese Rind Microbiome Disruption by a Spoilage Fungus)

Dear Prof. Laura M. Sanchez:

Please address the points raised by reviewer #1.

Revision Guidelines

Sincerely,
Trent Northen
Editor
mSystems

Reviewer #1 (Comments for the Author):

This manuscript describes the interaction between a filamentous fungus, *Aspergillus westerdijkiae*, and members of the cheese rind microbiome. The multi-omics nature of the manuscript provides a detail-orientated approach to identify the determinants of antagonistic interactions. The authors find that two mycotoxins, ochratoxin B and penicillic acid, are likely contributors to antagonism of *Brachybacterium*, whereas, they show little effect against *Staphylococcus*. I have no issues with any of the

methods or approaches, I think you have covered you bases.
Great work overall and it was a pleasure to read with some beautiful images.

I super happy that your medium included salt since you mention the brine solution is used on the cheese, I think you have mimicked the cheese environment as much as possible with your agar setup. It is interesting OTA is not observed when it is most often than not the predominant metabolite produced by this fungus. I think there could be something more to chase down here, is there a microbiome member that the *Aspergillus* does not want to inhibit (maybe for nutrients scavenging).

Two points I would like to make.

(1) Although you saw no affect of the mycotoxins on the *Staphylococcus*, it would have been a nice compliment to return back to a transcriptomics approach (or qPCR of the identified up-regulated genes in Figure 1E) and treat the Staph with pure mycotoxin. Although you do observe inhibition, at the sampling point of 4-days, there is not total inhibition. So could the mycotoxins (which are maybe still being produced at low concentrations) be mediating some affects on the *Staphylococcus* which are not bactericidal? Could you be seeing the increase of *bceAB* cluster because it is effluxing these mycotoxins? This could be one experiment that closed a small gap.

(2) is a pH question. We have found before that pH is a big indicator if fungal acids are produced or not. Fungi produce these as a principal way to allow for the excretion of enzymes to breakdown nutrients, so in already acidic environments (I didn't see any pH measurements in the methods). This pH effect could be dictating the interactions as well. So in Figure 6, you show that the interaction with *brachybacterium* increases the penicillic acid production. An alternative is that *brachybacterium* produces small bases (ammonia) to make the medium alkaline, and therefore to compensate, your *Aspergillus* then has to produce more acid to make nutrients available. I think you are safe in your workflows and conclusions, but this hypothesis could be answered by culturing with Bromocresol purple in the medium.

This manuscript by Grundmann *et al* describes the chemical fungal-bacterial interactions between *Aspergillus westerdijkiae*, a cheese spoilage mold, with the native bacteria from cheese rind. This work details mechanisms for how *A. westerdijkiae* can outcompete the native bacteria using key metabolites. Although the title focuses on penicillic acid, the manuscript identifies many more chemical signals that may play a role in these interactions. Overall, the data is very comprehensive and compelling. However, the manuscript would benefit from revision to create a clear throughline linking the various datasets together to tell a more integrated story. Additionally, improving the readability of the figures would aid in enabling the reader to follow the story.

- As a reader, it was difficult to follow the throughline of the story being told with the data. The inhibition data is with four bacteria and two fungi. The RNAseq work is from the interaction of *A. westerdijkiae* with *S. equorum*. The MSI data is from the interaction of *A. westerdijkiae* with *B. alimentarium*. The LC-MS/MS data is from the interactions of *A. westerdijkiae* with either *S. equorum* or *B. alimentarium*. And the inhibition assay with penicillic acid is with the four original bacteria. All the data is valid, but the justifications for each of the different experiments and how they relate to each other are unclear. Stronger explanations and context linking each of the experiments would be beneficial. For example, justification is provided by the switch to *B. alimentarium* for the MSI experiments, but it reads as an afterthought rather than considered experimental design.
- Many of the figures are difficult to interpret due to layout, small font, or lack of guidance for the reader to aid in data interpretation. The addition of jitters to the bar plots would also be helpful.
- To aid in interpretation of Figure 1, it would be beneficial to the reader if the conclusions that were summarized in the figure legend are moved to the main text and the portions for each panel are streamlined to guide the reader. For example, for (B) it would be helpful to the reader if the approach to measuring total CFUs as the sum of the CFUs from each microbe was stated rather than referring to panel (C). Additionally, panel (B) could be broken into two panels or a different symbol used for *Aspergillus* alone to help visually differentiate the community CFUs from the CFUs of only *Aspergillus*.
- In Figure 2, the ppm values for mass error are rather large considering the data was collected on a tims-FLEX. Due to the variation in sample height of the agar sample, the higher ppm values are understandable. However, in Figure 3, the ppm values for mass error are much smaller. Which raises the question whether the putative annotations in Figure 2 are accurate. Are the mass errors calculated from MSI data or LC-MS data for Figures 2 and 3?
- It is impossible to read the m/z values in panel (B) of Figure 4.

- If there are sufficient replicates, the data in Figure 5 should be quantified and represented as charts with statistical analyses rather than as XICs.
- The PCA/PLS-DA analyses are mentioned briefly, further discussion of the differential response of *A. westerdijkiae* to the two bacteria would strengthen the conclusion regarding customized responses to microbial partners.
- In Figure 6, are the peak areas in panel (A) relative abundances or can they be converted to absolute values as in panel (B)? It is unclear from the figure legend and main text how panel (A) data was quantified and how it relates to how the data in panel (B) was quantified.
- In the supplementary information, Figures S2 – S4 show mirror plots with an m/z range from 100 – 1800. The m/z values for these three metabolites are m/z 171, m/z 394, and m/z 292. The m/z range of the figures makes it impossible to visually compare the fragments between the library spectrum and the microbial samples. It would be helpful if the m/z range was adjusted to reflect the precursor m/z .

Editorial Comment:

Each figure must be uploaded as a separate, editable, high-resolution file (TIFF or EPS preferred), and any multipanel figures must be assembled into one file.

We have prepared each main text figure as a TIFF for upload.

Reviewer 1:

This manuscript describes the interaction between a filamentous fungus, *Aspergillus westerdijkiae*, and members of the cheese rind microbiome. The multi-omics nature of the manuscript provides a detail-orientated approach to identify the determinants of antagonistic interactions. The authors find that two mycotoxins, ochratoxin B and penicillic acid, are likely contributors to antagonism of *Brachybacterium*, whereas they show little effect against *Staphylococcus*. I have no issues with any of the methods or approaches, I think you have covered your bases. Great work overall and it was a pleasure to read with some beautiful images.

We thank the reviewer for their comments on our manuscript. We are excited that you enjoyed reading it alongside the figures!

I am super happy that your medium included salt since you mention the brine solution is used on the cheese, I think you have mimicked the cheese environment as much as possible with your agar setup. It is interesting OTA is not observed when it is most often than not the predominant metabolite produced by this fungus.

We agree that the lack of OTA production is unusual, we are aware that production of this chlorinated natural product does require salt, which we provide in the media. Other data involving the detection of OTA from cheeses have found that presence in hard rind cheeses (10.3390/toxins14050306), although co-occurrence with sterigmatocystin was also measured with STC being found in 94.4% of the samples compared to just 48.6% of the samples containing OTA. Further in a different sampling of artisanal cheese OTA was only found in 22% of the samples with *Aspergillus* section *Circumdati* producing OTA in every sample on Yeast Extract Sucrose Agar (YESA) at 25 °C/7 days, which is very different from our cheese curd media (10.1016/j.foodres.2024.114214). We posit that production of OTA is context dependent, but is beyond the scope of this manuscript since we have shown that penicillic acid is responsible for growth inhibition of specific microbes from our community.

All cheese rind microbial isolates that we have tested so far are inhibited to some extent by *A. westerdijkiae*. There are likely other members that may not be impacted but we have not comprehensively tested every cheese isolate. In Figure 1D we show that the yeast *Debaryomyces* is the least inhibited. We agree that this could be an interesting future direction that we are working towards, but is beyond the scope of this manuscript.

Two points I would like to make.

(1) Although you saw no effect of the mycotoxins on the *Staphylococcus*, it would have been a nice compliment to return back to a transcriptomics approach (or qPCR of the identified up-regulated genes in Figure 1E) and treat the Staph with pure mycotoxin. Although you do observe inhibition, at the sampling point of 4-days, there is not total inhibition. So could the mycotoxins (which are maybe still being produced at low concentrations) be mediating some affects on the *Staphylococcus* which are not bactericidal? Could you be seeing the increase of *bceAB* cluster because it is effluxing these mycotoxins? This could be one experiment that closed a small gap.

We agree that there may be some interesting non-bactericidal effects of the mycotoxins produced by the *Aspergillus* on the *Staphylococcus* species. We also appreciate the suggested experimental ideas that could help fill gaps in our mechanistic understanding of how bacteria respond to fungal metabolites in this system. However, the student who conducted this experiment has graduated and we no longer have the staff or funds to support additional transcriptomic experiments at this time. Even with the mechanistic gaps, this transcriptomic dataset is one small part of the overall storyline of this paper that demonstrates the production of specific metabolites by the fungus that can inhibit bacterial growth.

(2) is a pH question. We have found before that pH is a big indicator if fungal acids are produced or not. Fungi produce these as a principal way to allow for the excretion of enzymes to breakdown nutrients, so in already acidic environments (I didn't see any pH measurements in the methods). This pH effect could be dictating the interactions as well. So in Figure 6, you show that the interaction with *brachybacterium* increases the penicillic acid production. An alternative is that *brachybacterium* produces small bases (ammonia) to make the medium alkaline, and therefore to compensate, your *Aspergillus* then has to produce more acid to make nutrients available. I think you are safe in your workflows and conclusions, but this hypothesis could be answered by culturing with Bromocresol purple in the medium.

We appreciate the reviewer's insightful suggestion and performed a bromocresol purple pH-indicator assay to directly evaluate whether *Brachybacterium* alkalizes the medium during co-culture with *A. westerdijkiae*. Plates were imaged and analyzed using a semi-quantitative calibration curve based on extracted Blue/Red intensity ratios, allowing us to estimate pH values across defined regions of interest (ROIs) in addition to visual inspection of the indicator colors.

A**B**
Across all replicates, *A. westerdijkiae* consistently acidified the surrounding medium. The region adjacent to the fungal colony (ROI 1) displayed a strong yellow/pale-purple shift corresponding to an estimated pH of ~4.8 in both monoculture and co-culture. Importantly, *Brachy bacterium* did not produce an alkaline halo in any condition: the agar surrounding bacterial spots remained purple, and no blue-ish purple coloration indicative of ammonia-driven basification was observed.

In the interaction plates, a mildly acidic to near-neutral region (ROI 2) was detected at a further region of the fungal-bacterial interface where the fungal acidic halo gradually dissipated (pH ~6.1). We interpret this as a likely natural pH gradient where fungal acids disperse outward and are partially buffered by the medium and mild bacterial metabolic activity, rather than evidence of substantial base production by *Brachy bacterium*.

A key observation from this experiment is that the fungus is unequivocally acidifying the medium, and the pH pattern in co-culture does not resemble the spatial distribution of penicillic acid observed by MALDI-MSI. In MSI, penicillic acid is strongly and selectively induced at the fungal-bacterial interaction zone, whereas the pH gradient shows a smooth radial dilution rather than a localized hotspot. These spatial discrepancies indicate that acidification cannot be solely attributed to penicillic acid, and that pH shifts alone do not explain the interaction-specific induction of this metabolite.

Together, these results show that *Brachy bacterium* does not substantially basify the medium and therefore is unlikely to induce penicillic acid production via pH compensation. While local pH gradients are present, the pronounced increase in penicillic acid during co-culture is unlikely to be explained solely by pH effects and instead reflects partner-specific regulatory cues. We have briefly mentioned these observations to the Results section and added the figure in the Supplementary material (Fig. S7 and Table S3) accordingly.

Reviewer 2

This manuscript by Grundmann et al describes the chemical fungal-bacterial interactions between *Aspergillus westerdijkiae*, a cheese spoilage mold, with the native bacteria from cheese rind. This work details mechanisms for how *A. westerdijkiae* can outcompete the native bacteria using key metabolites. Although the title focuses on penicillic acid, the manuscript identifies many more chemical signals that may play a role in these interactions. Overall, the data is very comprehensive and compelling. However, the manuscript would benefit from revision to create a clear throughline linking the various datasets together to tell a more integrated story. Additionally, improving the readability of the figures would aid in enabling the reader to follow the story.

We thank the reviewer for their comments on our manuscript and find the data comprehensive and compelling!

1. As a reader, it was difficult to follow the throughline of the story being told with the data. The inhibition data is with four bacteria and two fungi. The RNAseq work is from the interaction of *A. westerdijkiae* with *S. equorum*. The MSI data is from the interaction of *A. westerdijkiae* with *B. alimentarium*. The LC-MS/MS data is from the interactions of *A. westerdijkiae* with either *S. equorum* or *B. alimentarium*. And the inhibition assay with penicillic acid is with the four original bacteria. All the data is valid, but the justifications for each of the different experiments and how they relate to each other are unclear. Stronger explanations and context linking each of the experiments would be beneficial. For example, justification is provided by the switch to *B. alimentarium* for the MSI experiments, but it reads as an afterthought rather than considered experimental design.

We appreciate this point. We tried to explain why we used particular microbes for particular assays in our original submission, but that may not have come through. So we added additional text to explain our rationale.

At the end of the introduction, we did some reorganizing and added text, including:

The specific microbial community modeled in this study is a washed-rind cheese, one type of artisan cheese where *A. westerdijkiae* has been observed directly disrupting rind formation (**Fig. 1A**). Washed-rind cheeses are repeatedly treated with a brine solution during aging to promote the development of a surface community dominated by bacteria and yeasts. Common bacterial genera include *Staphylococcus*, *Brevibacterium*, *Brachybacterium*, and various Proteobacteria, while *Debaryomyces hansenii* is the most prevalent yeast (4). Molds (filamentous fungi) are typically not abundant in washed rinds due to the frequent washing, although *Fusarium* (or *Bifusarium*) *domesticum* can persist despite this process.(3, 4) The presence of *A. westerdijkiae* in washed rind cheeses leads to the formation of patches where rinds do not develop and the surface of the cheese is white or brown instead of typical orange colors. This lack of typical rind development led us to hypothesize that *A. westerdijkiae* has antimicrobial activity that prevents beneficial rind microbes from growing.

We integrated MALDI-MSI spatial metabolomics with a variety of other approaches to comprehensively understand how *A. westerdijkiae* disrupts washed rind cheese rind development. First, we used interactions experiments with a model cheese rind community to determine what bacterial and fungal species are inhibited by *A. westerdijkiae*. Once we completed this interaction screen, we used transcriptome sequencing on one well studied focal bacterium, *Staphylococcus equorum*, to understand bacterial responses to *A. westerdijkiae*. This is a very well-studied food-associated *Staphylococcus* species with considerable past transcriptome studies, allowing us to compare transcriptional response to *A. westerdijkiae* to other interactions. Once we observed strong antibacterial growth and transcriptional responses, we then focused our assessment of the dynamics of metabolite production with *S. equorum* and another rind bacterium, *Brachybacterium alimentarium*.

At the start of the results section, we also added this text:

The experimental communities contained a full suite of bacteria and fungi that represent some of the dominant microbes found on washed rind cheeses: the bacteria *S. equorum* strain BC9 (Firmicutes), *Brevibacterium aurantiacum* strain BW86/JB5 (Actinobacteria), *B. alimentarium* strain BW87/JB7 (Actinobacteria), and *Psychrobacter* sp. strain BW96/JB193 (Proteobacteria); the yeast *Debaryomyces* sp. strain 135B; and the mold *Fusarium domesticum* strain 554A. In this initial set of community experiments, we included all typical community members, including both bacteria and fungi, to understand the full range of interactions between *A. westerdijkiae* and these microbes. In the mechanistic experiments that follow below (MALDI-MSI, LC-MS/MS, penicillic acid assays, etc.), we only focused on bacteria because of their dominance in washed rind cheese communities and because of their strong responses to *A. westerdijkiae*. Our mechanistic experiments below often focused on just *S. equorum* and *B. alimentarium*, for reasons explained in each section below. All experiments used a single *A. westerdijkiae* strain (Aw2019). and often just *S. equorum* and *B. alimentarium*, for reasons explained in each section below.

2. Many of the figures are difficult to interpret due to layout, small font, or lack of guidance for the reader to aid in data interpretation. The addition of jitters to the bar plots would also be helpful.

We appreciate this comment. Unfortunately, we are not sure exactly what specific figures the reviewer is referring to here. We have reviewed all of our figures carefully and have not found cases where a small font prevents reading, or where layout is confusing. Jitters can sometimes be helpful for understanding variation around a mean, but we do not believe they would add value to bar plots or change conclusions about the data we have presented. We appreciate the specific figure advice in #3 below (and have made those changes), but are unsure of other major figure issues that prevent interpretation by the reader.

3. To aid in interpretation of Figure 1, it would be beneficial to the reader if the conclusions that were summarized in the figure legend are moved to the main text and the portions for each panel are streamlined to guide the reader. For example, for (B) it would be helpful to

the reader if the approach to measuring total CFUs as the sum of the CFUs from each microbe was stated rather than referring to panel (C). Additionally, panel (B) could be broken into two panels or a different symbol used for *Aspergillus* alone to help visually differentiate the community CFUs from the CFUs of only *Aspergillus*.

We have removed some text from this legend to help add clarity. We have changed the symbol for the *Aspergillus* alone CFU counts as suggested.

4. In Figure 2, the ppm values for mass error are rather large considering the data was collected on a tims-FLEX. Due to the variation in sample height of the agar sample, the higher ppm values are understandable. However, in Figure 3, the ppm values for mass error are much smaller. Which raises the question whether the putative annotations in Figure 2 are accurate. Are the mass errors calculated from MSI data or LC-MS data for Figures 2 and 3?

We appreciate the reviewer's observation regarding the difference in mass error (ppm) values between Figures 2 and 3. This discrepancy is expected and is directly related to how the annotations were generated for each figure.

For Figure 2:

The putative annotations were first predicted through genome mining and subsequently verified for presence directly in the MALDI-MSI dataset. Therefore, the mass errors reported in Figure 2 come from the MSI acquisition itself. Because the agar substrate introduces variation in sample height and ionization efficiency as pointed out in this reviewer critique, the ppm values observed in this figure are inherently larger, even with a timsTOF fleX instrument.

For Figure 3:

In contrast, the annotations in Figure 3 were obtained through LC-MS/MS molecular networking, where mass accuracy is significantly higher due to stable chromatographic conditions and the absence of sample-height variation. The MSI data were used only to confirm spatial localization after the features had already been confidently annotated by LC-MS/MS. Thus, the ppm values reported in Figure 3 reflect the LC-MS mass errors, which are considerably smaller.

In summary:

- Figure 2 ppm values = MSI-derived, therefore larger due to agar height variation.
- Figure 3 ppm values = LC-MS/MS-derived, therefore smaller and more precise.
- MSI was used to confirm spatial presence, but mass accuracy in Figure 3 originates from LC-MS/MS data.

We have clarified this distinction in the revised manuscript in the legend of each figure.

5. It is impossible to read the m/z values in panel (B) of Figure 4.

In the revised version, we have increased the font size of all *m/z* and intensity labels, improved the contrast, and adjusted the layout to enhance legibility.

6. If there are sufficient replicates, the data in Figure 5 should be quantified and represented as charts with statistical analyses rather than as XICs.

We thank the reviewer for this suggestion. The experiment shown in **Figure 5** was designed as an exploratory, proof-of-detection assay rather than a quantitative comparison. Our goal at this stage was to confirm that the two key metabolites (compounds **1** and **6**) were produced under the different interaction conditions and to visualize how their *relative intensities* changed depending on the bacterial partner. This preliminary assay was also highly labor-intensive to perform, and its intention was specifically to guide the narrative by illustrating the qualitative trends that motivated the downstream targeted analysis.

The patterns observed in these XICs helped us formulate the hypothesis that *B. alimentarium* induces a stronger fungal metabolic response than *S. equorum*. This hypothesis was later tested and confirmed for compound **6** in a dedicated targeted LC-MS quantification experiment (**Fig. 6**), which is presented in the manuscript and provides the appropriate statistical assessment. Compound **1** was not subjected to quantitative analysis due to its lower detectability, and because compound **6** displayed a clearer and more biologically relevant secretion pattern as explained in the main text.

For these reasons, we believe that showing **Figure 5** as qualitative XICs is the most appropriate representation of this exploratory experiment and serves its intended role in the story: to illustrate the emergence of a species-specific fungal metabolic response that was later validated quantitatively.

7. The PCA/PLS-DA analyses are mentioned briefly, further discussion of the differential response of *A. westerdijkiae* to the two bacteria would strengthen the conclusion regarding customized responses to microbial partners.

In the original text, our intention was to highlight that multivariate analyses revealed distinct chemical phenotypes for each interaction, but we agree that this section would benefit from further clarification. We have now expanded the Results to explicitly describe how the PCA/PLS-DA scores plots show clear separation between the *A. westerdijkiae* × *B. alimentarium* and *A. westerdijkiae* × *S. equorum* co-cultures, indicating that the fungus adopts interaction-specific metabolic programs. We also detail the loadings that drive this separation, emphasizing that notoamides 4, 10, and 11 specifically characterize the response to *S. equorum*, whereas compound 6 is strongly associated with interaction with *B. alimentarium*. This expanded explanation more clearly links the multivariate statistics to the partner-specific metabolite profiles and strengthens the conclusion that *A. westerdijkiae* deploys customized chemical strategies depending on the interacting bacterial species. The updated paragraph discussing this analysis is below:

Since compounds **1** and **6** appeared to be more prominently produced during interaction with *B. alimentarium*, we next asked whether *A. westerdijkiae* deploys alternative chemical strategies when interacting with *S. equorum*. In this case, we performed multivariate analysis that included PCA and PLS-DA on the LC-MS/MS datasets from both co-cultures and the corresponding controls. The resulting scores plots revealed clear separation between the *A. westerdijkiae* × *B. alimentarium* and *A. westerdijkiae* × *S. equorum* interactions, indicating that the fungus adopts distinct chemical phenotypes in each context (**Fig. S8A–D**). Examination of the loadings showed that compound **6** strongly influenced the position of the *B. alimentarium* co-cultures, consistent with its prominent upregulation in this interaction. In contrast, the separation of the *S. equorum* co-cultures was driven by a different set of metabolites, notably notoamides **4**, **10**, and **11**, which were selectively enriched in this interaction but minimally detected with *B. alimentarium*. Together, these multivariate patterns demonstrate that *A. westerdijkiae* deploys a partner-specific chemical arsenal, adjusting its biosynthetic output depending on the identity of the competing bacterium.

8. In Figure 6, are the peak areas in panel (A) relative abundances or can they be converted to absolute values as in panel (B)? It is unclear from the figure legend and main text how panel (A) data was quantified and how it relates to how the data in panel (B) was quantified.

Thank you for pointing out the lack of clarity regarding the quantification shown in **Figure 6**. We have revised both the main text and the figure legend to clarify that the peak areas presented in panel (A) represent the raw LC-MS peak areas extracted from the EICs, which were subsequently used to calculate the absolute concentrations shown in panel (B). As described in the Methods, concentrations were determined by correlating these peak areas with the penicillic acid calibration curve generated using eight concentrations (0.1–25 µM) from PA commercial standard.

To improve clarity, we now explicitly indicate in the main text and in the figure legend that panel (A) shows peak areas prior to conversion and that panel (B) reflects the corresponding quantified values derived from the calibration curve. These changes should help readers understand how the data in panels (A) and (B) relate to one another as follows:

Main text:

Using LC-MS, we first extracted the peak areas for penicillic acid from the EICs in each sample and then converted these values to absolute concentrations using a standard calibration curve generated with purified penicillic acid (**Fig. S6**). Quantification was carried out for *A. westerdijkiae* monocultures, co-cultures with *B. alimentarium* and *S. equorum*, and the corresponding bacterial–bacterial interaction zones (BBI).

As shown in **Fig. 6A**, raw peak areas were markedly higher in *A. westerdijkiae*–*B. alimentarium* co-cultures relative to the fungal monoculture and the *S. equorum*

interaction. When these peak areas were converted to concentrations (**Fig. 6B**), co-cultures with *B. alimentarium* consistently showed the highest levels of penicillic acid (approximately 9.4 μM). In contrast, *A. westerdijkiae* monocultures and *S. equorum* co-cultures produced substantially lower concentrations (approximately 3.7 μM and 1.9 μM , respectively). We also detected penicillic acid in the BBI zone of the *Brachy bacterium* plates, though at lower concentrations than in the direct fungal–bacterial interface (FBI). This pattern suggests diffusion of the metabolite away from the interaction zone, diminishing with increasing distance from the interaction interface which is aligned with the MSI and ESI data (**Fig. 3** and **5**, respectively) shown before.

Figure Legend:

Figure 6. Comparison of penicillic acid peak area **and quantified concentrations** across different culture conditions. **(A) Peak areas obtained from extracted ion chromatograms (EICs) for penicillic acid (PA). These values represent the raw LC-MS peak areas used as input for quantification** in different experimental conditions: monoculture of *A. west.* (red), the interaction interface between *A. west.* and *Brachy bacterium* (dark green), the interaction interface between *A. west.* and *S. equorum* (dark blue), and the bacterial side of the interaction between *Brachy bacterium* × *Brachy bacterium* (light green) and *Staphylococcus* × *Staphylococcus* (light blue). Individual data points represent biological replicates ($n = 3$). **(B) Absolute concentrations of PA (μM) represented in the defined sampling zones on cheese agar plates, calculated by correlating the peak areas from panel (A) with the eight-point calibration curve generated using analytical standard (0.1–25 μM ; Fig. S6).** Values represent PA levels in the fungal control, fungal-bacterial interaction (FBI) and bacterial-bacterial interaction (BBI) zones (represented as dashed circles) across three plate conditions. Statistical analysis was performed using one-way ANOVA followed by Tukey's Honest Significant Difference (HSD) test for multiple comparisons. Statistically significant differences are indicated by asterisks (*, **, ***), where $p < 0.05$ (*), $p < 0.01$ (**), and $p < 0.001$ (***)

9. In the supplementary information, Figures S2 – S4 show mirror plots with an m/z range from 100 – 1800. The m/z values for these three metabolites are m/z 171, m/z 394, and m/z 292. The m/z range of the figures makes it impossible to visually compare the fragments between the library spectrum and the microbial samples. It would be helpful if the m/z range was adjusted to reflect the precursor m/z .

We have updated Figures S2–S4 by adjusting the m/z range of each mirror plot to center around the corresponding precursor ions (m/z 171, 394, and 292). The revised figures are now included in the Supplementary Information.

Re: mSystems01305-25R1 (**Spatial Metabolomics Reveals the Role of Penicillic Acid in Cheese Rind Microbiome Disruption by a Spoilage Fungus**)

Dear Prof. Laura M. Sanchez:

Your manuscript has been accepted, and I am forwarding it to the ASM production staff for publication. Your paper will first be checked to make sure all elements meet the technical requirements. ASM staff will contact you if anything needs to be revised before copyediting and production can begin. Otherwise, you will be notified when your proofs are ready to be viewed.

Sincerely,
Trent Northen
Editor
mSystems

Reviewer #1 (Comments for the Author):

I am completely satisfied with all changes and responses to my previous comments. I was really pleased the authors went beyond just responding and attempted to observe the pH differences in the agar. Your analysis was very well done and I think only helps define the mechanistic understanding you observed. Great work by all authors, this was a real pleasure to read after revision.

Reviewer #2 (Comments for the Author):

The authors have addressed all my comments. I appreciate their thoroughness and thoughtful responses.